# Dysfunctional natural killer cells can be reprogrammed to regain anti-tumor activity

Batel Sabag [ID][1,2], Abhishek Puthenveetil [ID][1,2], Moria Levy [ID][1], Noah Joseph[1], Tirtza Doniger [ID][1], Orly Yaron[1], Sarit Karako-Lampert[1], Itay Lazar[1], Fatima Awwad [ID][1], Shahar Ashkenazi [ID][1] & Mira Barda-Saad [ID][1✉]

## Abstract

**Natural killer (NK) cells are critical to the innate immune system, as they recognize antigens without prior sensitization, and contribute to the control and clearance of viral infections and cancer. However, a significant proportion of NK cells in mice and humans do not express classical inhibitory receptors during their education process and are rendered naturally "anergic", i.e., exhibiting reduced effector functions. The molecular events leading to NK cell anergy as well as their relation to those underlying NK cell exhaustion that arises from overstimulation in chronic conditions, remain unknown. Here, we characterize the "anergic" phenotype and demonstrate functional, transcriptional, and phenotypic similarities to the "exhausted" state in tumor-infiltrating NK cells. Furthermore, we identify zinc finger transcription factor Egr2 and diacylglycerol kinase DGKα as common negative regulators controlling NK cell dysfunction. Finally, experiments in a 3D organotypic spheroid culture model and an in vivo tumor model suggest that a nanoparticle-based delivery platform can reprogram these dysfunctional natural killer cell populations in their native microenvironment. This approach may become clinically relevant for the development of novel anti-tumor immunotherapeutic strategies.**

**Keywords** NK Cell Anergy; NK Cell Immunotherapy; Egr2; DGKα; Molecular Mechanisms of NK Cell Dysfunction
**Subject Categories** Cancer; Immunology

## Introduction

Natural killer (NK) cells are lymphocytes of the innate immune system, providing the first line of immunosurveillance against viral infections and tumor growth. Most human NK cells express inhibitory receptors, such as the killer cell immunoglobulin-like receptors (KIRs) in humans and the heterodimeric inhibitory receptor CD94-NKG2A (hereafter referred to as NKG2A) in humans and mice, all of which recognize major histocompatibility complex class-I (MHC-I) molecules. The immune system employs intricate regulatory mechanisms to ensure that immune cells distinguish foreign invaders from healthy tissues. NK cells eliminate target cells lacking the expression of MHC-I molecules, whereas MHC-I-expressing cells are unaffected by NK cells, accounting for NK cell tolerance. In this regard, NK cells are mainly tuned by target cells expressing MHC-I molecules in their surroundings (Long et al, 2013). This process of self-tuning, calibrated through "education" allows NK cells to acquire functional competence and host-specific adaptations (Raulet and Vance, 2006; Kim et al, 2005; Orr and Lanier, 2010). Overall, NK cell education is determined by coordinating inhibitory, activating, and adhesion signals, through which inhibition and activation of NK cells are functionally linked (He and Tian, 2017). This allows NK cells to assess the balance of activating versus inhibitory signals they receive.

The NK education models, including the licensing, disarming, rheostat, and confining models, propose that classical killing inhibitory receptors (KIR) play an instructive role in NK cell responsiveness (Raulet and Vance, 2006; Kim et al, 2005; Anfossi et al, 2006; Boudreau and Hsu, 2018; Fernandez et al, 2005; Long et al, 2013; Zhang et al, 2019; Chen et al, 2016a; Wu et al, 2016; Brodin et al, 2009). These "instructive" inhibitory receptors contain immune-receptor tyrosine-based inhibitory motifs (ITIM) in their cytoplasmic tails, with subsequent recruitment and activation of protein tyrosine phosphatases (PTP), including the Src homology region-2 domain (SH2)-containing phosphatase 1/2 (SHP-1/2), or SH2 domain-containing inositol polyphosphate 5-phosphatase 1 (SHIP1), following binding to MHC-I. Recently, it was shown that SHP-1 expression levels and NK cell functional responsiveness are tightly linked, suggesting the involvement of SHP-1 in the molecular control of the rheostat determining NK cell responsiveness (Wu et al, 2021). However, the molecular mechanisms underlying the control of NK cell responsiveness remain elusive. Due to the diversity in the affinity and the amount of self-MHC-I inhibitory receptors among NK cells, the strength of the educating signal differs from cell to cell (Brodin et al, 2009). Signaling mediated by inhibitory receptors, such as NKG2A and KIR, acts primarily at the early stages of Immunological synapse (IS) formation, to abolish the activating signals (Long et al, 2013). Following education, NK cell reactivity increases with the number of different self-MHC-I-specific inhibitory receptors expressed (Thomas et al, 2013; Jaeger and Vivier, 2012). If the target cell lacks

[1]The Mina and Everard Goodman Faculty of Life Sciences, Bar-Ilan University, Ramat-Gan 5290002, Israel. [2]These authors contributed equally: Batel Sabag, Abhishek Puthenveetil. ✉E-mail: Mira.Barda-Saad@biu.ac.il

MHC-I or expresses low levels of this surface marker, it results in NK cell activation and clearance of the target. However, failure to engage inhibitory receptors during development, due to lack of inhibitory receptor expression on the NK cell or lack of interaction with MHC-I, results in the generation of a subset of non-responsive peripheral NK cells termed "anergic cells" (Kim et al, 2005; Fernandez et al, 2005).

"Anergy" describes a state in which the NK cell is intrinsically functionally impaired (Joncker et al, 2010). It was suggested that anergy might be an induced state, resulting from chronic exposure of NK cells to activating ligands without proper inhibitory signaling (Brodin et al, 2009; Tripathy et al, 2008). On the other hand, in the context of cancer, inhibitory receptor signaling destabilizes the IS, and promotes NK cell detachment and migration (Burshtyn et al, 2000). Upon encountering target cells expressing activating ligands but lacking MHC-I, NK cells are highly activated. However, if persistent, the over-activation of NK cells leads to their desensitization abolishing further interactions with additional targets such as cancer cells, leading to an "exhausted" NK cell state (Jaeger and Vivier, 2012).

Accordingly, NK cell dysfunction reflects different states, including anergy and exhaustion, each with a distinct etiology. "Anergic" NK cells are naturally unresponsive peripheral blood cells, constituting ~13 ± 6% of the entire NK cell population (Raulet et al, 2003; Fernandez et al, 2005). They lack expression of MHC-I-specific inhibitory receptors, and are not autoreactive, but rather tolerant to self (Anfossi et al, 2006; Fernandez et al, 2005). In contrast, "exhausted" NK cells arise from chronic viral infections, inflammation, or cancer, because of their overstimulation by their targets. Deciphering the molecular events leading to such a state and understanding the commonalities between these two populations is essential for their reprogramming to fight infections and cancer.

NK cells play a complementary role to T cells in tumor immunity by recognizing tumors that downregulate MHC-I expression and escape CD8+ T cell-mediated tumor clearance (Fruci et al, 2013; Lanier, 2008). Although the anti-tumor role of NK cells has been described in hematological malignancies, their role in the solid tumor milieu remains unclear due to their lack of activity in the tumor microenvironment (TME) (Guerra et al, 2008; Kreisel et al, 2012; Sanchez et al, 2011). Clinical observations highlight NK cells as a critical component in the anti-tumor response (Nersesian et al, 2021; Remark et al, 2013; Eckl et al, 2012), but despite tumor infiltration, a functional NK-mediated anti-tumor response is often lacking in the TME (Childs and Carlsten, 2015; Schleinitz et al, 2010; Kwon et al, 2017).

While significant progress has been made in understanding the molecular mechanisms of T-cell dysfunction, the equivalent pathways remain relatively unexplored for NK cells. Despite their clinical relevance, the heterogeneity, molecular, and transcriptomic landscape underlying NK cell dysfunction remain poorly defined. Furthermore, it is unclear if NK cell anergy and exhaustion reflect separate and distinct mechanisms of dysfunction, although both states are typically characterized by decreased effector function or proliferation (Judge et al, 2020). Therefore, our study aims to unravel the molecular mechanisms and etiology that drive NK cell dysfunction, providing insights for manipulating these cell populations to enhance anti-tumor responses.

Here, we identify key intrinsic checkpoints in NK "anergy", including diacylglycerol kinase (DGK)-α and early growth response (Egr)-2, and reveal functional, phenotypic and transcriptional similarity with NK cell "exhaustion". We further demonstrate in situ reprogramming of "anergic" and "exhausted" NK cells by modulation of these key intrinsic regulators, DGKα and Egr2, via a nanoparticle (NP)-based drug delivery platform (Biber et al, 2021), both in vitro and in vivo, revealing that the identified markers are critical to reprogram both the dysfunctional states. In particular, our in vivo model of NK cell exhaustion reveals that gene silencing of Egr2 empowers NK cells to effectively control tumor growth and enhance NK cell effector functions for improved tumor lysis and clearance. Furthermore, our data suggest that "anergy" and "exhaustion" are not simply intrinsic non-responsive states, but that these newly identified targets can potentially enable these dysfunctional NK cells to be reprogrammed in their native environment, to become functional in diverse contexts, including cancer and viral infections.

## Results

### Transcriptomic profiling and analysis of anergic NK cells

To characterize the "anergic" NK cell population, we first obtained NK cells from the peripheral blood of healthy individuals and sorted, via flow cytometry, the anergic NK cell subset, lacking self-MHC-I-specific inhibitory receptors (pan KIR and NKG2A) and the responsive NK cell subset, which expresses the complete inhibitory receptor repertoire (Anfossi et al, 2006; Fernandez et al, 2005) (Fig. EV1A). We next confirmed that the isolated NK populations are each functionally anergic or responsive by performing degranulation and tumor lysis assays (Fig. EV1B,C). To address the transcriptional landscape of these populations, we performed RNA-seq analyses of the isolated anergic vs. responsive sub-population. Unbiased principal component analysis (PCA) results showed a clear separation between these populations along principal component (PC) 2 (variance of 81%), illustrating the high degree of transcriptomic differences (Fig. 1A). The top significant genes belonging to the anergic, and responsive populations were filtered and gene expression differences in each donor were quantified by hierarchical clustering (Figs. 1B and EV1D). The obtained heatmap visually illustrates the extent of variability among differentially expressed genes within anergic and responsive subsets of NK cells. Notably, minimal transcriptional variations exist between donors within the same subset (Fig. 1B). The enriched pathways for each NK subset were analyzed by Metascape (Zhou et al, 2019) (Fig. EV1D–G). The network analysis of the top-hit significant genes and the associated pathways showed a highly interconnected circuitry governing the molecular framework of NK cell responses (Fig. EV1F).

To infer regulatory interactions between human transcriptions factors (TFs) and target genes within the anergic subset, TRRUST analysis using the Metascape tool was performed on the RNA-seq dataset (Fig. EV1E), revealing the enrichment of several immune effector-related TFs within the anergic subset. Notably, these included NFkB and RELA (p65 subunit of NFkB), which play critical roles in regulating inflammatory responses and in various aspects of immune regulation (Ronin et al, 2019; Oeckinghaus and Ghosh, 2009); JUN, responsible for AP-1 complex formation; and SATB1, a TF showing functional relevance in immune regulation and tumorigenesis (Sunkara et al, 2018).

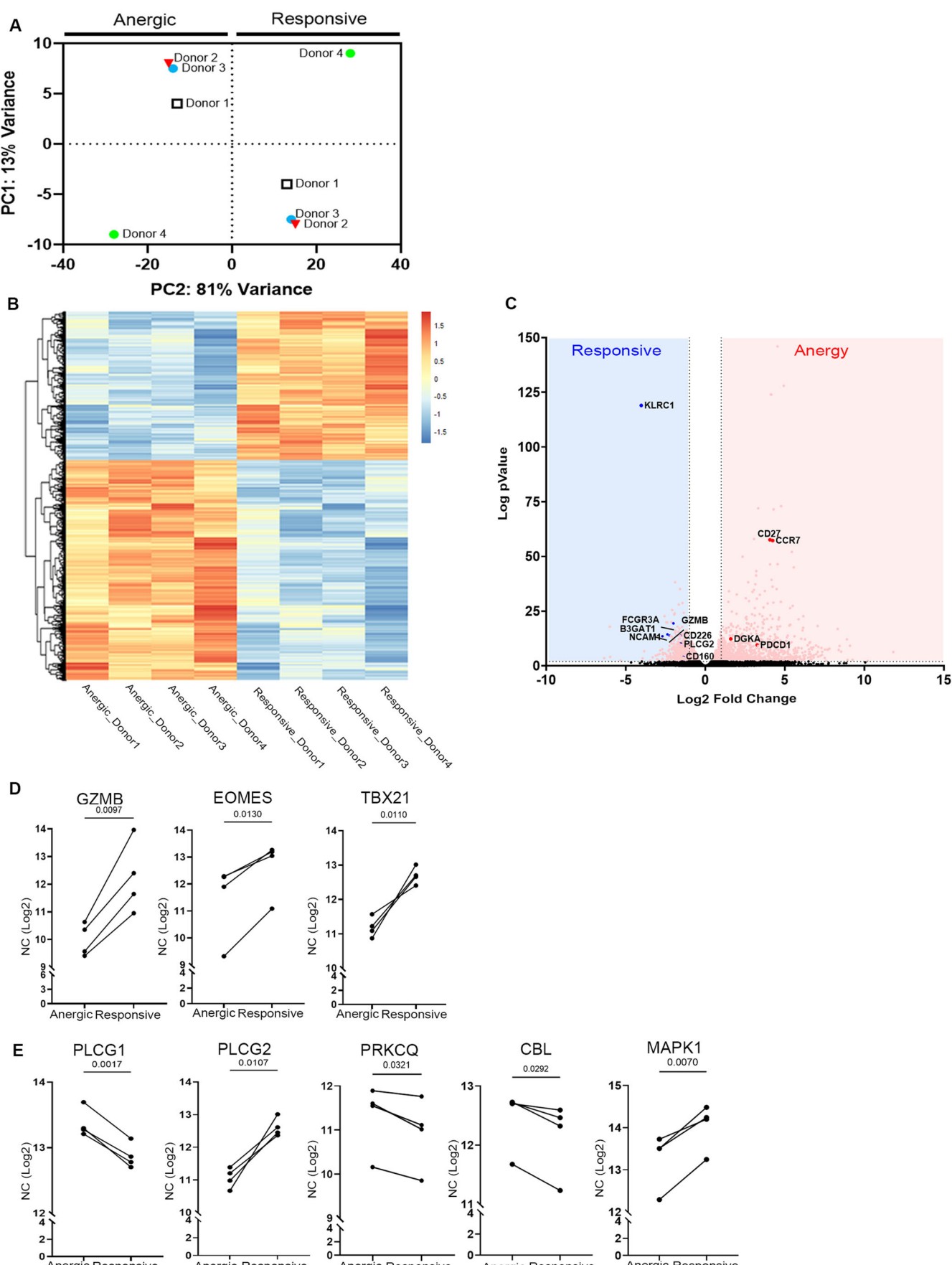

**Figure 1. Gene expression profiling and analysis of anergic and responsive populations.**

(A) PCA performed using DESeq2 on the anergic vs. responsive subset vst normalized RNA-seq data. The variance is depicted on the $X$ and $Y$ axes, and each individual donor is indicated. The anergic and responsive populations demonstrate high transcriptional differences reflected by a variance of 81%. (B) Heatmaps (transcripts per kilobase million normalization with Z score) showing expression of representative genes in anergic vs responsive populations from individual donors. The heatmaps illustrate subtle distinctions among the donors with noteworthy similarities. (C) Volcano plots showing differentially expressed genes in anergic vs. responsive NK cells ($n = 4$ healthy donors used to isolate pNK and sort anergic and responsive NK cell populations). Selected differentially expressed genes with an adjusted $P$ value $\leq 0.05$ and log2 fold change >1 or −1 are indicated in the plot. Gene-specific $t$ test was used for statistical analysis. (D, E) The transcript levels of key genes expressed in the anergic, and responsive populations were normalized to log2 and are presented as line dot graphs. The $P$ value was calculated using a two-paired tailed $t$ test and is indicated within the graph (NC normalized counts). Genes in (D) reflect the markers of effector and functional maturity of NK cells—*GZMB, Eomes, TBX21,* and those in (E) represent the transcript levels of key signaling molecules involved in NK cell responses, *PLCG1, PLCG2, PRKCQ, CBL,* and *MAPK1.* The respective $P$ values are indicated in the graph, and each line represents the values of a single donor ($n = 4$ healthy donors). Source data are available online for this figure.

A volcano plot was further generated to help identify significantly upregulated genes with high fold change in the anergic subset. One such gene is *CD27* which serves as a maturation marker for the NK cell lineage, with the CD27[high] NK cell subset exhibiting a distinct tissue distribution and responsiveness to chemokines and interacting productively with dendritic cells (Hayakawa and Smyth, 2006). The increased transcript level (TL) of *CCR7* in anergic cells was intriguing, as a previous study reported that CCR7 expression in NK cells improves migration toward lymphomas and helps in tumor control (Schomer et al, 2022). Taken together, the obtained results indicate the potential of the anergic NK cell subset in immune regulation (Fig. 1C). In addition, the upregulation of *KLRC1* (NKG2A) in the responsive NK population supports the positive role of NKG2A in acquiring functional competence, reflecting its significant role in NK cell education and functional responses (Kim et al, 2005). Moreover, its expression in responsive cells not only validates the sorting procedure and its efficiency but also confirms that the anergic population is devoid of NKG2A expression. Furthermore, the upregulation of genes associated with NK cell function, such as *FCGR3A* (encoding CD16), *NCAM1* (encoding CD56), *GZMB* (encoding Granzyme B), *CD226* (encoding DNAM-1), *PLCγ* (encoding PLC gamma 1/2), *B3GAT1* (encoding CD57), and *CD160* (encoding CD160) in responsive cells indicates the compromised functionality of the anergic NK cells.

The "anergic" population was previously shown to exhibit diminished effector functions (Ardolino et al, 2014) with the underlying signaling pathways unknown. To investigate the basis of the impaired functionality, we focused on examining the transcript levels of key genes known to contribute to effector functions and key genes playing a major role in the signaling cascades mediating NK cell cytotoxicity (Fig. 1D,E). The dysfunction of the anergic NK population was reflected by the reduced TL of *Eomes* and *Tbx21*, which are TFs known to play a role in the maturation of NK cells (Kiekens et al, 2021), suggesting compromised maturation and reduced ability of the anergic population to control tumor growth or infections. Analysis of the TL of crucial signaling molecules associated with NK cell activation was conducted in anergic and responsive subsets (Fig. 1E); notably, we observed significant reductions in PLCγ2 and MAPK TL in anergic cells, collectively implying compromised functionality within the anergic population.

## Naturally induced "anergic" NK cells display a transcriptional program analogous to that of "exhausted" NK/T cells, exhibiting an overlap of crucial regulators

We next examined whether the anergic NK cell subset exhibited alterations at the receptor level. The expression of the key activating receptors, *FCGR3A* (encoding CD16) and *NCR1* (encoding

NKp46), which determine NK cell activity (Moretta and Moretta, 2004; Bryceson et al, 2006), were found to be significantly reduced at both the RNA and protein level in anergic cells (Fig. 2A). In addition, TL and surface expression of CD160, an important biomarker for cytokine production in NK cells and early control of tumor growth (Tu et al, 2015) was found to be reduced in anergic cells along with the TL and surface expression of DNAM-1 (Du et al, 2018) and 2B4 (Meazza et al, 2017), which are vital receptors responsible for NK cell responses (Fuchs and Colonna, 2006; Enqvist et al, 2015) (Fig. 2B). These transcriptional changes indicate that NK cell anergy is characterized by downregulation of key activating surface receptors. Furthermore, it is noteworthy that these anergic NK cells do not express any inhibitory surface markers such as NKG2A or classical KIR's. However, the TL of *PDCD1*, the gene encoding PD-1 was found to be significantly elevated within the anergic subset, and TLs of *HAVCR2* (encoding Tim-3) and *TIGIT* (encoding TIGIT) were lower. Next, we determined and confirmed these at the protein level via flow cytometry (Fig. 2C). The data thus suggest that PD-1 could potentially serve as a marker for identifying anergic NK cells. The combined transcriptional and protein data reveal a unique signature of signaling molecules playing a role in NK cell anergy.

The distinctive attributes including downregulated activating receptors and compromised signaling pathways within anergic NK cells, prompted us to explore the potential concurrence of transcriptional and phenotypic characteristics between this naturally induced "anergic" phenotype and the NK cell "exhausted" phenotype, typically observed in the context of viral infections or cancer. To this end, gene set enrichment analysis (GSEA) was conducted, comparing our designated gene sets to gene expression datasets from NK cells obtained from patients with chronic Hepatitis B virus (HBV) infection vs. NK cells from healthy donors (Marotel et al, 2021). The gene sets characteristic of NK cell anergy exhibited enrichment towards the top of the ranked list, including genes obtained from individuals with HBV infections, notably *DGKA*. This observation elucidates a pool of genes shared between NK cell anergy and NK cells sourced from viral infection (Fig. 2D).

In addition, we sought to understand whether anergic NK cells share a similar signature with T cells during viral infection and cancer with a focus on identifying key highly enriched genes. To address this, our dataset was compared to datasets from (West et al, 2011) and revealed transcriptional commonalities of the anergic NK cells with virally exhausted T cells (Fig. 2E), highlighting noteworthy genes including *DGKA, EGR2,* and *NR4A2*. In addition, we explored whether the anergic NK gene profile overlapped with the canonical "CD8[+] T-cell exhaustion" program by obtaining gene sets from Martinez et al (NFAT signature) (Martinez et al, 2015a)

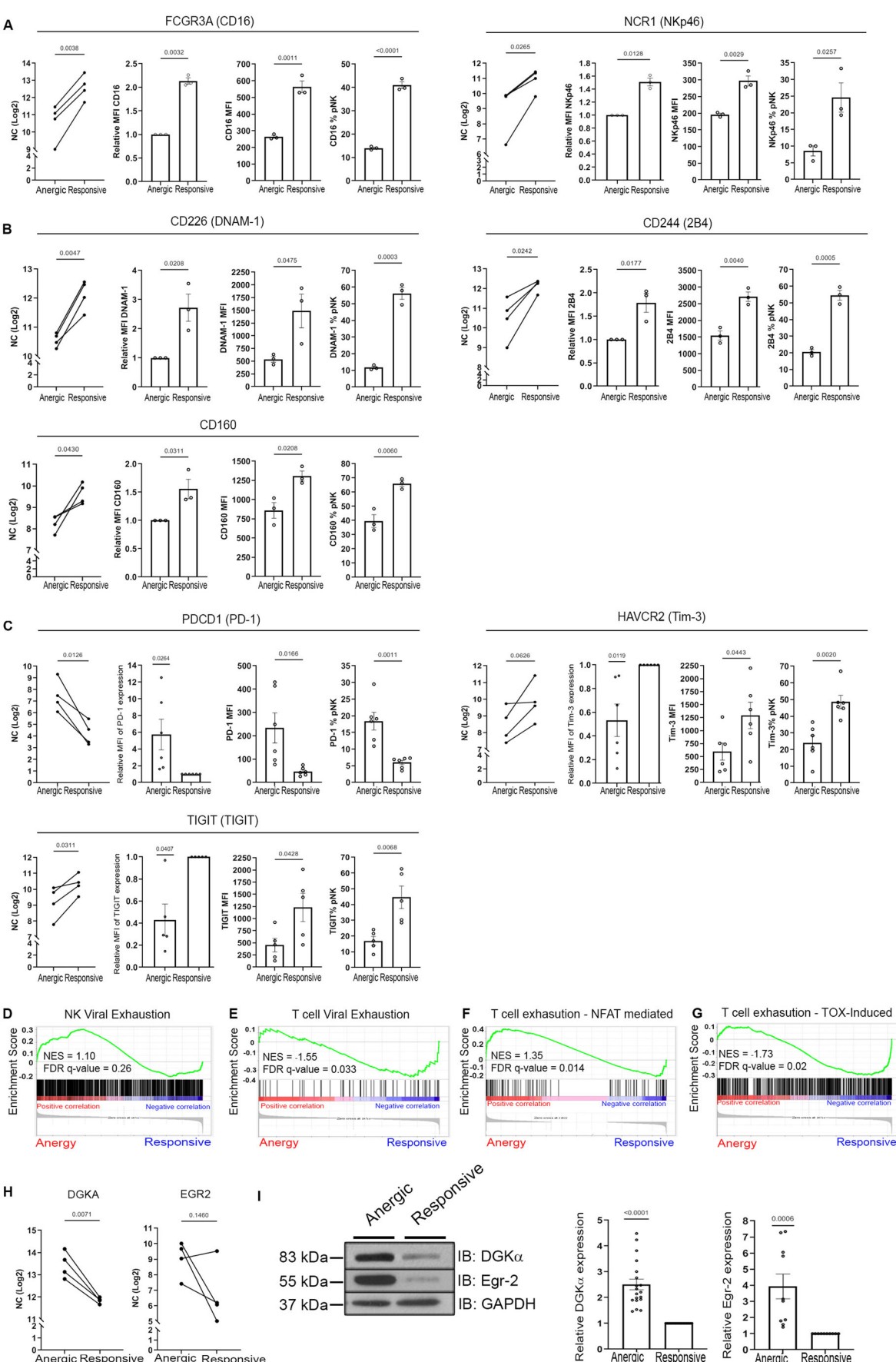

Figure 2. Naturally induced "anergic" NK cells share a transcriptional program similar to "exhausted" NK/T cells with an overlap of key regulators.

(A) The transcript and protein levels of key genes reflecting NK cell activation expressed in anergic and responsive populations. RNA levels were normalized to log2 and are presented as line graphs (RNA), or bar graphs (protein- flow cytometry); *FCGR3A* (CD16), *NCR1* (NKp46) ($n = 4$ donors for the RNA and for the FACS data, $n = 3$ healthy donors). The *P* value was calculated using a two-tailed paired *t* test and is indicated within the graph. The panels represent (from left to right): RNA transcript normalized count, relative MFI, raw MFI and percentage of parent. (B) Transcript and protein levels of DNAM-1, 2B4, and CD160. Each line represents a single donor for RNA ($n = 4$ healthy donors), and dots represent individual donors for protein. The *P* value was calculated using a two-tailed paired *t* test and is indicated above the graph. For each marker, panels represent (from left to right): RNA transcript normalized count, relative MFI, raw MFI, and percentage of parent. (C) The transcript levels of key inhibitory surface checkpoint markers, *PDCD1, HAVCR2*, and *TIGIT* ($n = 4$, NK cells sourced from four healthy donors for RNA seq, left). Protein expression of the key inhibitory surface checkpoint markers was obtained from the number ($n$) of healthy donors thus mentioned, PD-1 ($n = 6$), Tim-3 ($n = 6$), and TIGIT ($n = 5$). The panels represent (from left to right): RNA transcript normalized count, relative MFI, raw MFI, and percentage of parent. (D) Gene Sequence Enrichment Analysis (GSEA) using gene sets from exhausted NK cells obtained from HBV-infected individuals vs healthy donors compared to the anergic and responsive population (Marotel et al, 2021). The anergic NK cells showed positive correlation to the HBV-exhausted NK cells, whereas responsive NK cells showed positive correlation to the NK cells from healthy donors; NES value = 1.10, FDR *q* value = 0.26. (E) GSEA using gene sets from datasets of viral CD8 T-cell exhaustion (West et al, 2011); NES value = −1.55, FDR *q* value = 0.033. (F) NFAT-induced T-cell exhaustion datasets from (Martinez et al, 2015b). NES = 1.35, FDR *q* value = 0.014 (G) HMG BOX mobility factors, TOX and TOX2 induced T-cell exhaustion datasets from (Scott et al, 2019b). NES = −1.73, FDR *q* value = 0.02. The statistical tests for the GSEA (D–G) were performed using the weighted Kolmogorov–Smirnov test. (H) Transcript levels of key genes identified from the GSEA, *DGKA, EGR2* ($n = 4$, NK cells sourced from four healthy donors for RNA seq). (I) Purified responsive and anergic NK cells from healthy donors ($n$) were lysed and subjected to western blot with anti-DGKα ($n = 18$) and anti-Egr2 ($n = 10$) antibodies (left panel). Graphs showing the densitometric quantitative values of the western blots, with each dot representing an experimental repeat (right panel). Data are presented as mean ± SEM. *P* values were calculated using a two-tailed *t* test with matched data repeats and are indicated within the graph. Source data are available online for this figure.

and Scott et al (TOX signature) (Scott et al, 2019a) (Fig. 2F,G). Interestingly, anergic NK cells demonstrated a significant upregulation of key genes associated with the T-cell exhaustion program, exhibiting substantial enrichment in prominent genes such as *PDCD1, NR4A1, NR4A3, EGR2, NFATc1*, and *TOX2*, aligning with both the NFAT- and TOX-signatures. To validate these signatures, we examined the expression of NFAT2, NFAT1, TOX, and TOX2 at the TL and NFATc2, NFATc1 expression at the protein level (Fig. EV2A,B). NFAT2 (NFATc1) and TOX2 were significantly upregulated in anergic cells, whereas NFAT1 (NFATc2) levels were reduced, and no significant change was observed in TOX levels between anergic and responsive cells (Fig. EV2C). Collectively, our data highlight the overlap of key significant genes such as *DGKA* (NK and T-cell viral infection), *EGR2* (T-cell viral infection and CD8 T-cell exhaustion), and the potential role of key TFs including *EGR2, NFAT2 (NFATc1)*, and *TOX2* in driving NK cell anergy.

## An in vivo model substantiating shared intrinsic regulators governing "exhausted" and "anergic" NK cells

The transcriptional and phenotypic similarities observed between naturally induced "anergic" NK cells and "exhausted" immune cell phenotypes in cancer and viral infections suggest a convergence of molecular pathways underlying immune cell dysfunction. High enrichment scores for overlapping genes such as Egr2, DGKα and DGKζ were observed in Fig. 2. Egr2 is a zinc finger TF known to have a role in regulating immune cell functions (Wagle et al, 2021; Dai et al, 2020). DGKα and DGKζ are enzymes playing a crucial role in lipid signaling and regulation of immune cell function via facilitating the conversion of Diacylglycerol (DAG) to phosphatidic acid (PA) (Zhong et al, 2008).

The high abundance of DGKα and EGR2 in anergic NK cells was confirmed at the RNA and protein levels (Fig. 2H,I). In addition, anergic NK cells exhibited elevated DGKζ protein expression (Fig. EV2D). This implies that these intrinsic regulators contribute to the establishment of the intricate molecular circuit governing NK cell anergy.

To further investigate the similarities between "anergic" and "exhausted" NK cells, we sought to determine whether the NK cells

isolated from tumors exhibit a functional and phenotypic dysfunctional signature similar to naturally existing anergic NK cells. To address this, a xenogeneic tumor-grafted mouse model of pancreatic cancer was established. Cells from an aggressive human pancreatic ductal carcinoma (PANC-1) were introduced sub-cutaneously (*s.c.*) into NOD/SCID IL-2Rγ^null (NRG) mice. Primary human NK cells obtained from healthy donors were injected intra-tumorally (*i.t.*) (Fig. 3A). The administered NK cells were retrieved from the tumors when the decrease in tumor size reached a plateau at day 18, signifying the inability of the NK cells to further control tumor growth (Fig. 3B). These tumor-infiltrating NK cells (TINK) (identified based on hCD45+ expression (Fig. EV3A,B)) obtained from excised tumors on day 18 exhibited significantly reduced CD107a levels (Figs. 3C and EV3B,C), coupled with drastically upregulated PD-1 expression relative to the naive NK, indicating an "exhausted" phenotype (Fig. 3D). Furthermore, as anticipated, Egr2 and DGKα levels were significantly elevated in the TINK cells relative to the naive NK cells (Figs. 3E,F and EV3B). This in vivo system thus reveals striking functional and phenotypic similarity between "anergic" NK cells and TINK, reflecting an "exhausted" phenotype at the protein level, marked by reduced effector functions and drastically upregulated PD-1, DGKα, and Egr2 expression.

Altogether, our in vivo data demonstrate that the TINKs exhibit a dysfunctional phenotype reminiscent of the anergic NK cells, highlighting the potential therapeutic significance of targeting these intrinsic regulators. Therefore, inhibition of these identified key intrinsic regulators can not only potentially improve the responses of dysfunctional NK cells, but also the entire tumor-infiltrating lymphocyte (TIL) population, which is of immense therapeutic potential in immune regulation.

## Gene silencing of intrinsic NK regulators reverses NK cell dysfunction

Transcriptome profiling and protein expression data complemented with in vivo validation revealed underlying intrinsic regulators of NK activity. We therefore initially tested in vitro whether the gene silencing of DGKα, DGKζ or EGR2 in the anergic subset dictates the activation status and functional outcome of anergic NK

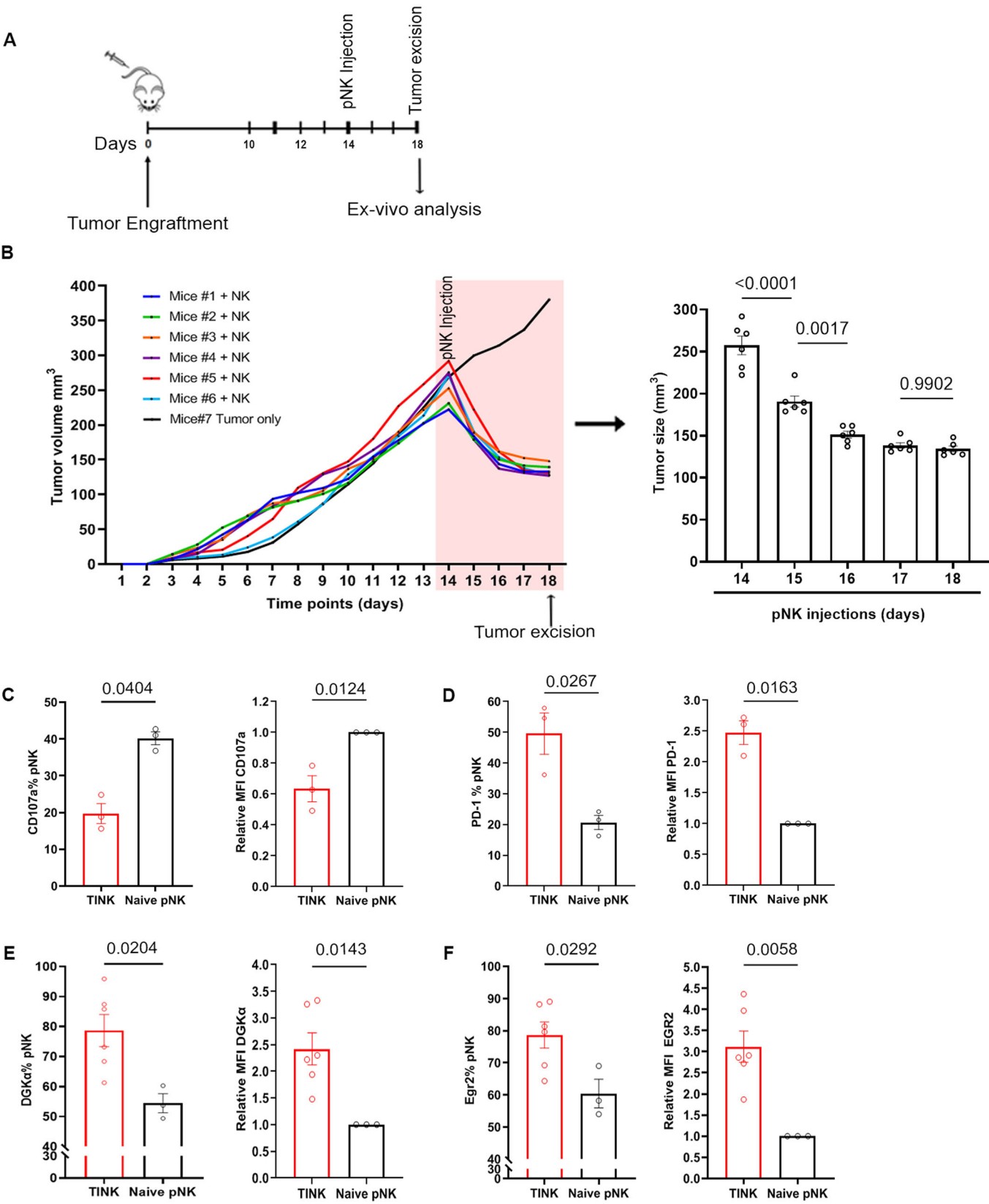

◀ **Figure 3. In vivo model exhibits shared intrinsic regulators governing tumor infiltrating "exhausted" NK cells and those controlling NK cell "anergy".**

(A) Experimental protocol. Timeline of in vivo experiment depicting tumor engraftment, effector administration, tumor excision, and ex vivo analysis. NOD/SCID IL-2Rγ$^{null}$ (NRG) mice engrafted with $4 \times 10^6$ aggressive human pancreatic ductal carcinoma cells (PANC-1) received a single infusion of primary human NK cells (*i.t.*). They were allowed to grow until the tumor size plateaued, indicating decreased ability of the NK to control the tumors. Tumor burden was assessed daily. (B) Left panel: Graph illustrating progression of tumor growth (*n* = 6 mice). On day 14 these mice received single intratumor (*i.t.*) infusion of $1.1*10^7$ human pNK from two healthy donors. The pink-shaded region on the graph corresponds to the period during which pNK cells were present within the tumor post injection (with exception of tumor-only group). The groups treated with pNK cells exhibited a significant reduction in tumor size, although this effect was short-lived, lasting only 4 days. This suggests a decrease in the tumor lysis capacity of the pNK cells after 4–5 days. Right panel: Graph based on data in left panel showing days 14–18 of the tumor growth and plateauing of the tumor size toward day 18 (experimental end point), suggesting the escape of the tumor cells from NK-mediated control. *P* values were calculated using one-way ANOVA with Tukeys post hoc multiple comparison test and are indicated within the graph. (C, D) FACS analysis: Tumors excised on day 18 were dissociated to single-cell suspensions. NK cells were subjected to flow cytometry and were differentiated by hCD45 expression. Representative graphs showing: Left panel: Percentage of pNK expressing CD107a and PD-1 (*n* = 3 mice) in naive vs TINK; Right panel: Relative MFI corresponding to the respective percentage values. (E, F) FACS analysis: Tumors excised on day 18 were dissociated to single-cell suspensions. NK cells were subjected to flow cytometry and were differentiated by hCD45 expression. Representative graphs showing: Left panel: Percentage of pNK expressing Egr2 and DGKα (*n* = 6 mice) in naive vs TINK; Right panel: Relative MFI corresponding to the respective percentage values. The *P* values were calculated using a two-tailed *t* test with pairing and are indicated within the graph (mean ± SEM) along with the number of experimental repeats. Each symbol represents an individual mouse. Source data are available online for this figure.

cells. As an initial step, isolated anergic and responsive cells were treated with DGKi (R59022) or DMSO (vehicle) and co-incubated with 721.221 target cells. Phospho-ERK Y204 (pERK) levels were evaluated by flow cytometry (Fig. 4A) and were found to be about fourfold lower in the anergic versus the responsive cells. However, following DGKi treatment, a significant increase in the phosphoERK levels of the anergic NK cells was observed, reaching levels observed in the responsive subset (Fig. 4A). Accordingly, recovered pERK expression levels reflect the restoration of mitogen-activated protein kinase (MAPK) activity in anergic cells following DGKi treatment. In addition, the anergic and responsive subsets were transfected with DGKα, ζ siRNA, or NS siRNA (Fig. EV4A). Anergic NK cells exhibited an approximately two-fold increase in degranulation following DGKα, ζ gene silencing (Fig. 4B), restoring activity of the anergic cells to levels similar to the responsive ones and confirming that DGKα and ζ play a major role in dictating NK cell responses.

Next, the anergic and responsive cells were treated with Egr2 siRNA or NS siRNA. Gene-silencing of Egr2 significantly recovered the cytotoxic potential of the anergic cells to a greater extent than that of DGKα, ζ siRNA, reaching degranulation similar to that of the responsive population (Figs. 4C and EV4B). To determine if the restored degranulation of anergic cells via gene silencing of EGR2 was DGKα-dependent, DGKα and Egr2 expression levels were quantified. The silencing efficacy of Egr2 was determined to be 55%, resulting in an expression level similar to that observed in responsive cells (Fig. 4D). DGKα expression was examined on the same blot to determine the effect of Egr2 silencing. As expected, we observed a decrease in DGKα expression in EGR2-silenced cells with a highly significant reduction of nearly 60%, similar to the DGKα expression observed in NS siRNA-treated responsive cells (Fig. 4D). These results collectively indicate that inhibition of Egr2 consequently decreases DGKα expression, eventually leading to overall functional reinvigoration of the anergic NK cells. Moreover, this validates EGR2-DGKα-MAPK-pERK as a critical signaling axis compromised in anergic NK cells.

Mechanistically, high DGKα and low pERK expression level in anergic cells (Fig. 4A) suggests reduced DAG availability and elevated levels of PA (Purow, 2015), resulting in SHP-1 recruitment to the cell membrane and enhanced activity (Frank et al, 1999). To this end, phosphorylation levels of SHP-1, PLCγ1 and PLCγ2 were measured to determine their activity in anergic cells. Low phosphoSHP-1 (S591) indicative of high SHP-1 activity (Ben-

Shmuel et al, 2022), and decreased pPLCγ1 (Y783) and pPLCγ2 (Y1217) levels were observed, reflecting compromised activity (Matalon et al, 2016) (Fig. EV4C–E). We next investigated if Egr2 knockdown influences SHP-1 activity. As expected, Egr2 gene silencing induced a significant elevation of phospho-SHP-1 (S591) levels in anergic NK cells, reflecting the inactivated form of SHP-1 and its reduced activity, similar to that in responsive NK cells (Fig. 4E).

The high SHP-1 activity in anergic NK cells suggests inhibition of SHP-1 substrates such as PLCγ1/LAT (Matalon et al, 2016). This compromises both the PLCγ mediated DAG and IP$_3$ signaling axes. Calcium levels mediated by IP$_3$ also serve as a hallmark for cell activity and responses through the calcineurin-activated NFAT pathway (Khan et al, 1992; Akimzhanov and Boehning, 2012). Since Egr2 silencing demonstrated a reduction in SHP-1 activity, Egr2 inhibition will exert an effect on the IP$_3$ signaling axis. Indeed, Egr2 gene silencing resulted in the restoration of calcium levels in anergic NK cells to levels almost comparable to those in responsive cells (Fig. 4F). These data indicate that Egr2 governs both the DAG and IP$_3$ signaling branches, reflected by reduced DGKα, and increased pERK, degranulation, and calcium levels. Thus, inhibiting Egr2 could potentially recover both signaling axes, and ultimately restore NK cell function.

As demonstrated above, PD-1 was significantly elevated in anergic cells. Moreover, effector functions negatively correlate with surface inhibitory receptor expression. Thus, to investigate the impact of EGR2 silencing on the expression of inhibitory checkpoint receptors, we performed EGR2 gene silencing in anergic vs responsive cells. We specifically observed a significant reduction in the expression levels of PD-1 (Fig. 4G) but no differences were observed in Tim-3 and TIGIT expression levels (Figs. 4H,I and EV4F).

Thus, our data collectively show that EGR2 silencing rescues cytotoxic potential via the EGR2-DGKα-MAPK-pERK axis, normalizing SHP-1 activity, calcium levels and reducing PD-1 expression, supporting its crucial role in NK cell dysfunction and as a potential therapeutic target.

## Reinvigorating dysfunctional NK cells in situ in 3D OTS and in vivo models via an NP delivery platform

We next investigated whether the DGKα and Egr2 NK cell signature observed in tumors from patients shows any correlation

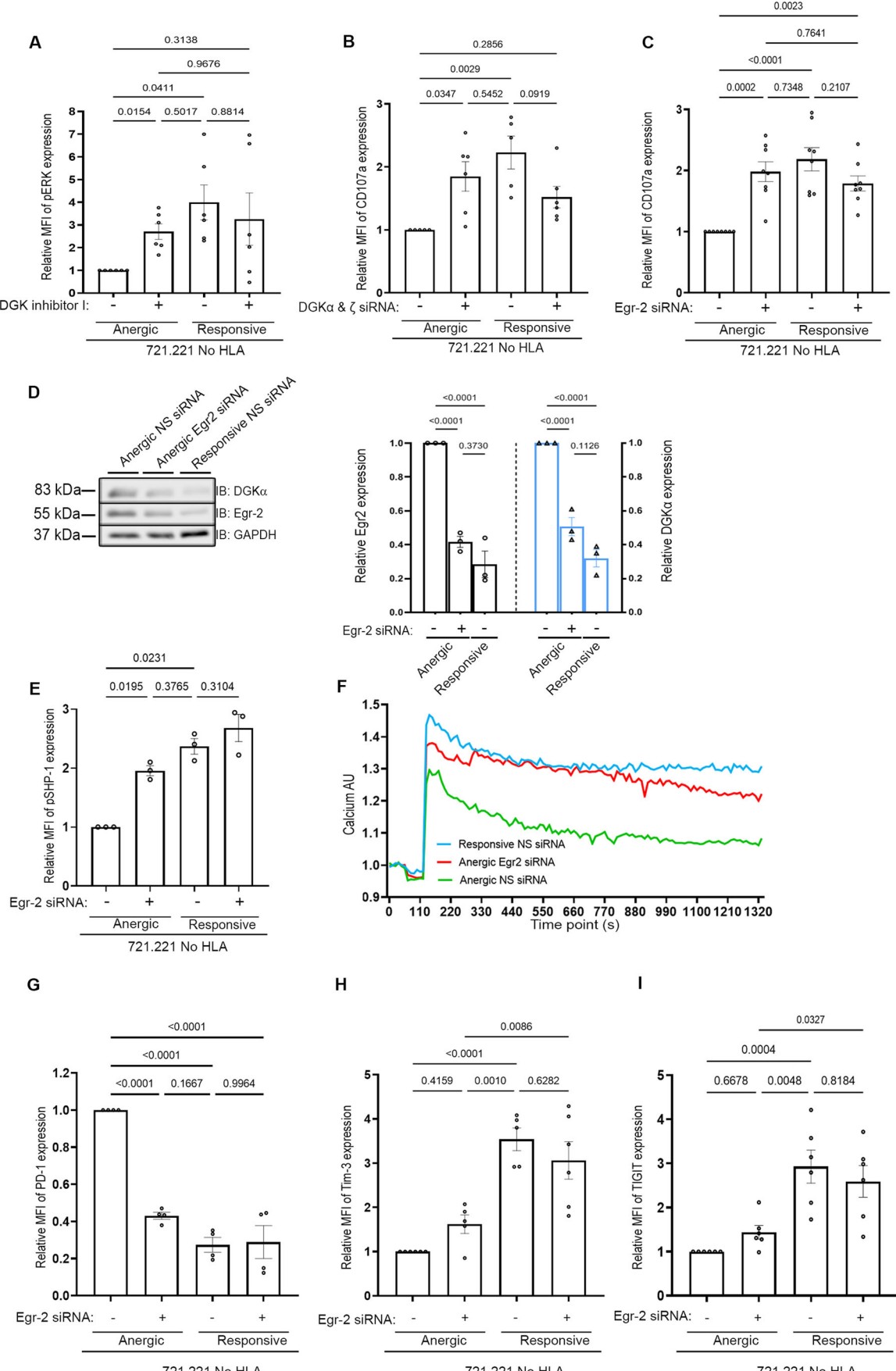

◄ **Figure 4. Knockdown of the intrinsic regulators reveals underlying molecular mechanism to reverse NK cell dysfunction.**

(A) The indicated pNK cell subsets were either incubated with D5919, a pharmacological inhibitor of DGK, or DMSO. The cells were co-incubated with 721.221 HLA-negative target cells for 5 h at 37 °C. The graph shows the normalized pERK levels analyzed by intracellular staining from six independent experiments. Data are presented as means ± SEM. *P* value was calculated using one-way ANOVA and Tukeys' post hoc test relative to the anergic NS treatment, and each experimental repeat is indicated by a dot ($n = 6$ healthy donors). The indicated pNK cell subsets were either transfected with (B) DGKα, ζ siRNA ($n = 6$ obtained from three healthy donors), or (C) Egr2 siRNA ($n = 10$ obtained from 7 healthy donors) or NS siRNA as a control following incubation with 721.221 HLA-negative target cells for 5 h at 37 °C. Degranulation was determined by measuring the MFI of CD107a-positive NK cells by FACS. The NK cells were distinguished from the target cells according to FSC and SSC. Graph summarizing the normalized CD107a-positive cells from eight and six independent experiments, respectively. Data are presented as means ± SEM. *P* value was calculated using one-way ANOVA relative to the anergic cells treated with DMSO/NS siRNA, and multiple comparisons were performed by Tukeys' post hoc test and are indicated within the graph. (D) Purified anergic and responsive NK cells treated with Egr2 or NS siRNA and subjected to lysis and western blot analysis with Egr2 and DGKα antibodies on the same membrane (left panel). Graph summarizing data representative of three independent experiments ($n = 3$ healthy donors) (right panel). Data are presented as means ± SEM. *P* value was calculated using one-way ANOVA, and multiple comparisons were performed by Tukeys' post hoc test and are indicated within the graph. (E) Purified anergic and responsive NK cells treated with Egr2 siRNA or NS siRNA as a control following incubation with 721.221 HLA-negative target cells for 5 h at 37 °C. Phospho(p)SHP-1 levels were measured via intracellular staining, and cells were subjected to flow cytometric analysis. The NK cells were distinguished from the target cells according to FSC and SSC ($n = 3$ healthy donors). Data are presented as means ± SEM. *P* value was calculated using one-way ANOVA relative to the anergic cells treated with DMSO/NS siRNA, and multiple comparisons were performed by Tukeys' post hoc test and are indicated within the graph. (F) Anergic cells were treated with either Egr2 or NS siRNA, and responsive cells were treated with NS siRNA. The cells were stained with Indo-1 AM, activated by PMA/ionomycin activation cocktail, and intracellular calcium flux was measured as the ratio between the $Ca^{2+}$ bound/unbound over 22 min. One experiment presented a representative of three (included in the source data file). The traces represent each condition, green line: anergic NS siRNA; red line: anergic Egr2 siRNA; and blue line: responsive NS siRNA. (G–I) Isolated human pNK cell subsets were either transfected with Egr2 siRNA or NS siRNA. Surface expression of inhibitory checkpoints receptors was obtained from the number ($n$) of healthy donor thus mentioned., PD-1 ($n = 6$), Tim-3 ($n = 5$), and TIGIT ($n = 6$). Graph summarizing the normalized APC-positive cells from independent experiments. Data are presented as means ± SEM. The *P* value was calculated using one-way ANOVA with a Tukeys' post hoc test, and is indicated within the graph, along with each experimental repeat. Source data are available online for this figure.

to patient survival. Indeed, NK cell infiltration and function are subjects of key interest in various cancer types including primary glioma, therapy-resistant glioblastoma (Breznik et al, 2022; Shaim et al, 2021; Sedgwick et al, 2020), and myeloid leukemia (Kaweme and Zhou, 2021; Carlsten and Järås, 2019). However, dysfunction of NK cells and impairment in their activity have been reported in these cancers. To determine whether the phenotype we observed correlates to the exhausted population observed in these tumors, we obtained the Genome Cancer Atlas datasets of acute myeloid leukemia and glioma. We found expression of NK cell-associated genes (*NCR1, NCR3, CD160, PRF1, KLRB1* (hereby referred to as the NK signature) in these cancer datasets (Böttcher et al, 2018; Ni et al, 2020). To this end, the NK cell signature was established along with *DGKA-EGR2*^high and *DGKA-EGR2*^low profile of samples in these cancer datasets (Fig. 5A,B). The NK-DGKA-*EGR2*^high signature correlated with significantly reduced survival of both leukemia and glioma. In addition, the NK cells were highly enriched in the immune cell infiltration profile, strengthening the contention that this signature belongs to the infiltrating NK cells. Overall, these results depict the immense therapeutic potential of targeting these key intrinsic regulators to improve cancer immunotherapy responses by restoring the functionality of hypo-responsive lymphocytes.

To determine the potential of NK cell activation by targeting EGR2, we analyzed the cytolysis of tumor cells by anergic NK cells gene silenced for EGR2. Anergic cells (transfected with Egr2 siRNA or NS siRNA) or responsive cells (transfected with NS siRNA) were tested for lysis of 721.221.HLA-Cw7 target cells expressing mCherry using the Incucyte assay. Indeed, we observed poor tumor cell lysis by the anergic cells treated with NS siRNA compared to the responsive population (Fig. EV5A; Fig. S1A) reflected by the decrease in fluorescence intensity from the mCherry target cells. Following EGR2 silencing, tumor lysis improved dramatically to levels similar to the cytolysis displayed by the responsive cells (Fig. EV5A). These results indicate that Egr2 indeed regulates anergy-associated genes, and its silencing reversed the functional dormancy of these NK cells.

To evaluate and validate potential therapeutic platforms, we established a "3D organotypic-spheroid (OTS) model" of human chronic myelogenous leukemia (CML) with K562- target cells expressing cyan fluorescent protein (CFP) with the aim of reprogramming anergic NK cells in situ. A liposomal nanoparticle (NP)-based siRNA delivery system, incorporating anti-NKp46 for targeting NK cells that was previously established in our lab based on SHP-1, Cbl-b, and c-Cbl siRNA to overcome NK cell inhibition (Biber et al, 2021), was used here for Egr2 siRNA transfer. We previously showed that K562 cells do not take up the anti-NKp46 tagged NPs, hence eliminating the possibility of off-target effects (Biber et al, 2021). To test the ability of Egr2 siRNA-loaded NPs to activate NK activity, we encapsulated the NP with Egr2 siRNA or NS siRNA (control) and incubated the NPs with the OTS cultures. Target cell lysis was measured over 48 h. Poor tumor cell lysis was attained by the anergic cells prior to anti-NKp46-tagged NP treatment, whereas responsive cells exhibited greater tumor cell lysis, as measured at 6 h. The anergic NK cells were then treated with Egr2 siRNA encapsulated NPs. They exhibited significantly greater target lysis than anergic cells treated with NS siRNA-NP (Figs. 5C and EV5B) reflected by a decrease in the target cell fluorescence intensity. Following NP-based Egr2 siRNA delivery at 48 h, cytotoxicity of the anergic cells was strongly enhanced, almost to the level of cytolysis displayed by the responsive cells (Figs. 5C and EV5B). Moreover, at 48 h, tumor lysis was similar in anergic cells treated with Egr2 siRNA encapsulated NP and responsive cells treated with NS siRNA-NP (Fig. EV5B). This provides preliminary proof of concept for use of a liposomal drug delivery platform to reprogram anergic and exhausted NK cells in situ to activate their anti-tumor activity.

Our current study demonstrates that the naturally 'anergic' NK cell population shares similar phenotypic, transcriptional, and functional characteristics with the "exhausted" NK cells. Therefore, we examined the ability of the exhausted NK cells to control tumor growth upon EGR2 gene silencing utilizing the NRG xenograft PDAC model (Fig. 3). Mice bearing PANC-1 tumors were injected *i.t.* with pNK when the tumors reached 250–300 mm³. Previously,

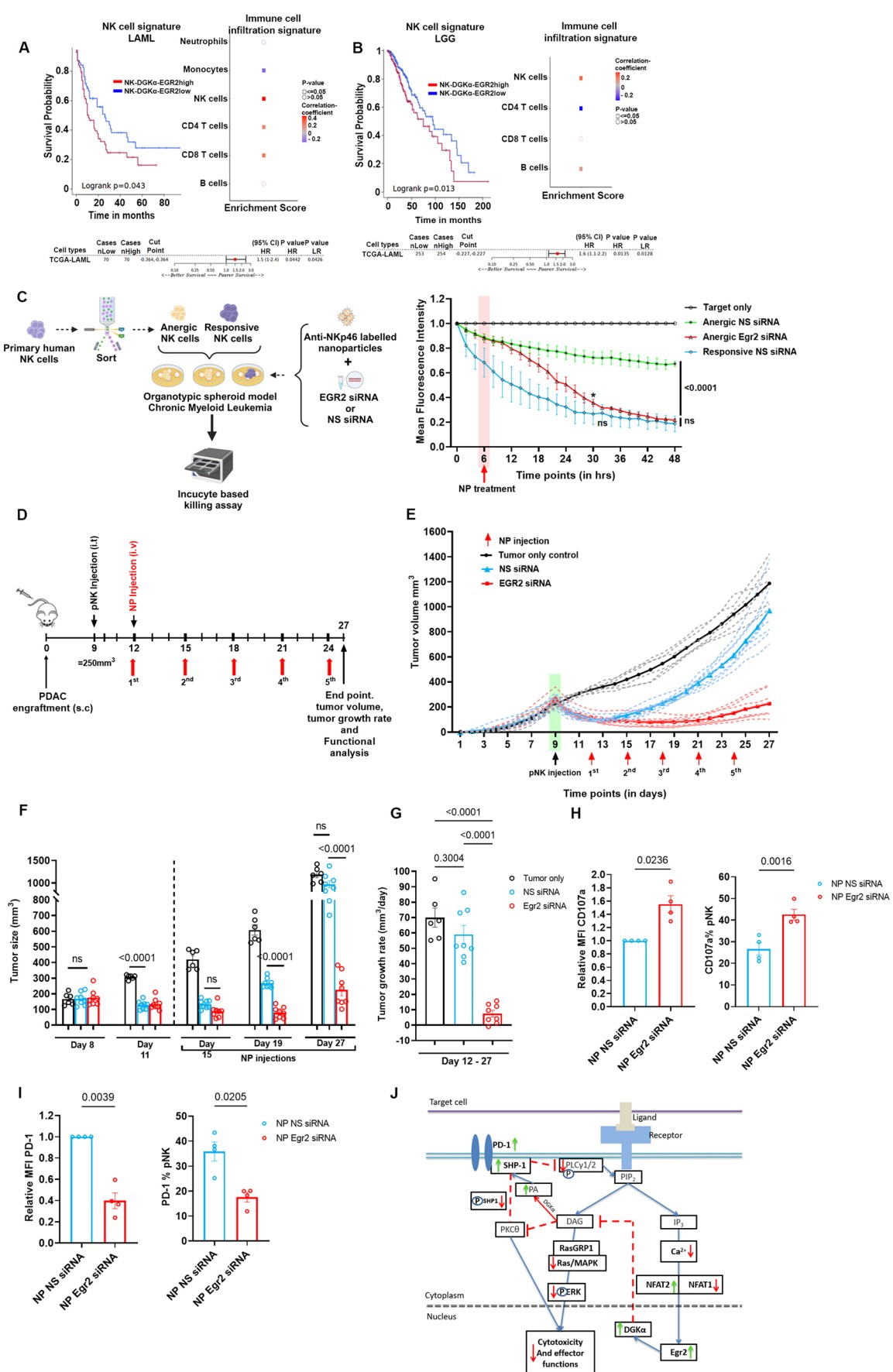

**Figure 5. Targeting Egr2 in situ using 3D organotypic spheroid (OTS) culture and in vivo model as a potential therapeutic approach.**

(A, B) Correlation of NK-*DGKA-EGR2* signatures with the overall survival of (A) AML patients, and (B) glioma patients comparing high and low quartiles. Kaplan Meier curves are presented showing patient survival (obtained from Survival Genie (Dwivedi et al, 2022)) along with the *P* values. Right panels show the signature of immune cell infiltration in AML and glioma; and red squares represent positive correlation to immune cells observed in the cancer dataset. Bottom panels indicate the correlation of the survival score to the NK cell signatures. *P* values (A, B) were calculated using Log-rank *t* test. (C) Left panel: Schematic representation of the 3D OTS model using the NP delivery platform to reprogram anergic NK cells in a tumor milieu. Primary NK cells were used to obtain anergic and responsive subsets and were then seeded to pre-constructed 3D cultures and administered with NP encapsulating Egr2 or NS siRNA. Next, the pNKs were subjected to an Incucyte-based killing assay. Right panel: Human chronic myeloid leukemia OTS 3D domes were established using Matrigel and cultured for 48 h in OTS media consisting of RPMI media supplemented with 1 µg/mL fibroblast growth factor (FGF), 0.18 µg/mL epidermal growth factor (EGF) and 500 IU/mL transforming growth factor (TGFβ). After 48 h, the 3D cultures were incubated with either freshly isolated anergic or responsive NK cells. After 6 h of NK-tumor co-incubation, NPs encapsulating Egr2 siRNA or NS siRNA were added (represented by the pink-shaded region). The decrease in fluorescence intensity reflects target cell lysis by the respective NK cell population and the associated treatment. The analysis was performed for 25 individual field frames for each experimental condition (with approx. $n = 30$ cells each field). Statistical analysis was conducted for three independent experiments, and presented as means ± SEM ($n = 3$ healthy donors). *P* value was calculated using two-way ANOVA with a Tukey's post hoc test for multiple comparison, and paired *t* test was performed between each time point and group to find significant changes in tumor lysis between the groups at specific time points. Anergic NS siRNA vs Anergic Egr2 siRNA (*P* value (*) = 0.0121, time point: 30 h). (D) Timeline of the in vivo experiment. PDAC-1-xenograft NRG mice were established as previously described and received a single infusion of $1.1 \times 10^7$ human pNK from four healthy donors on day 9 once the tumors reached ~250–300 cm²; "tumor only" control did not receive NK cells but received NS siRNA encapsulated NPs. NPs encapsulating Egr2 siRNA or NS siRNA were administered i.v. from day 12 for every 3 days until day 27. (E) Tumor volume (in mm³) was monitored and measured daily throughout the experiment. The pNK injection is depicted by a black arrow in a green-shaded region, and the NP injections are indicated by red arrows. Mice in the "tumor only" group ($n = 6$ mice) received no pNK treatment but were administered NP encapsulated NS siRNA. Mice that were administered with Egr2 siRNA ($n = 8$ mice) are depicted in red, while those receiving NS siRNA ($n = 8$ mice) are depicted in blue. The bold lines indicate the average, and the dashed lines represent individual mice within their respective experimental groups. (F) Tumor sizes (mm³) measured during the indicated days. Black graph represents the control group with tumor only. The blue and the red graphs represent groups of mice that received treatment with NP encapsulating NS siRNA or Egr2 siRNA, respectively. Tumor sizes at specific time points are indicated: Day 8 (one day before pNK injection) and Day 11 (2 days after pNK injection) are tumor sizes prior to NP injection; day 15 (6 days after pNK injection), day 19 (10 days after pNK injection) and day 27 (18 days after pNK injection and final day before tumor excision) represent tumor sizes following NP injection. Data are presented as mean ± SEM. *P* values are calculated using one-way ANOVA accompanied by a Tukeys' post hoc multiple comparison test individually for each day presented and are indicated within the graph. (G) Tumor growth rate measured from day 12 (first NP administration) until day 27 (end point) shown for the three groups (Egr2 siRNA-NP vs NS siRNA-NP vs tumor only). Data are presented as mean ± SEM. *P* values are calculated using one-way ANOVA and are indicated within the graph ($N = 4$ healthy donors were used to obtain the pNK; Groups – tumor only ($n = 6$), NP Egr2 siRNA ($n = 8$), NP NS siRNA ($n = 8$), where n is the number of mice). (H, I) Ex vivo analysis. The tumors were excised on day 27 and processed to single-cell suspensions by dissociation as described in the Materials and Methods. They were then stained for (H) CD107a ($n = 3$, where n is the number of mice used to obtain the pNK cells) and (I) PD-1 ($n = 3$, where *n* is the number of mice used to obtain the pNK cells). The pNK were distinguished based on hCD45 expression and NP incorporation (PE positive). Fluorescence is represented as both relative MFI (left panels) and percentage of pNK (right panel). *P* values were calculated using a two-tailed paired *t* test and are represented within the graph presented as means ± SEM. (J) Scheme depicting the proposed signaling pathway of anergic cells in accordance with the transcriptome and protein level profiling; PA phosphatidic acid, PLC phospholipase Cγ1/2, DAG diacylglycerol, DGK diacylglycerol kinase, Egr early growth response, MAPK mitogen-activated protein kinase, NFAT nuclear factor of activated T cells, PD-1 programmed cell death protein 1, PKCθ protein kinase Cθ, SHP-1 Src homology 2 domain-containing protein tyrosine phosphatase 1, pS591 phospho – S591. "Anergic" cells exhibit elevated EGR2 expression, which subsequently triggers an increase in DGKα. This leads to the conversion of DAG to PA, in turn, PA recruits more SHP-1 to the cellular membrane. As a result availability of DAG is restricted, which hampers PKCθ activity, rendering it incapable of modulating SHP-1 activity (Ben-Shmuel et al, 2022). This allows SHP-1 to dephosphorylate LAT and PLCγ1/2 (Matalon et al, 2016), preventing the initiation of a secondary cascade. DAG depletion also inhibits the activation of the DAG-mediated Ras-Raf-MEK-ERK pathway and IP3-mediated calcium flux. Consequently, there is no nuclear translocation of NFAT and its effector partners, such as AP-1, to initiate an effector response. This ultimately results in establishing an "anergy-associated gene transcription program". Source data are available online for this figure.

we observed upregulation of DGKα, Egr2, and PD-1 and reduced degranulation by day 5 following pNK administration (Fig. 3B–F). Accordingly, the mice were treated *i.v.* with NPs encapsulating EGR2 siRNA or NS siRNA 3 days following NK cell transfer, every 3 days for five treatments (Fig. 5D,E). A significant reduction of tumor volume and growth rate was observed in mice treated with EGR2 siRNA-NP vs NS siRNA-NP (Fig. 5F,G). To confirm that the differences observed in the tumor size and growth rate were related to the NK cell activity, the tumors were excised on Day 27 and subjected to ex vivo analyses. Degranulation was significantly higher in NK cells obtained from the mice that received NP EGR2 siRNA treatment vs NP NS siRNA (Fig. 5H). In addition, PD-1 expression was decreased in EGR2 siRNA-treated cells (Fig. 5I), as was the expression of Egr2 itself (Fig. EV5C). Furthermore, the tumors extracted from the group treated with EGR2 siRNA displayed increased apoptosis compared to the group treated with NS siRNA. Collectively, the in vivo data demonstrate that EGR2 siRNA-NP-treated NK cells exhibit superior anti-tumor activity over the control group, NS siRNA-NP-treated NK cells.

Our results show that elevated expression of DGKα and EGR2 correlates with reduced patient survival. We further demonstrate that a liposomal nanoparticle-based siRNA delivery system

effectively reprogrammed anergic NK cells within a 3D organotypic spheroid leukemia model. In addition, in xenograft models of pancreatic cancer, EGR2 gene silencing enhanced the anti-tumor potency of exhausted NK cells, leading to augmented tumor control, reduced tumor growth, decreased PD-1 expression, and the restoration of the cytotoxic potential of exhausted NK cells. In summary, our study unravels the potential molecular circuitry of anergic and exhausted NK cells and demonstrates Egr2-DGKα-SHP-1-PLCγ1/2-MAPKas a critical signaling axis in NK cell dysfunction (as illustrated in Fig. 5J).

## Discussion

The modulation of NK cell responsiveness represents a promising approach in cancer therapy due to their innate ability to target cancer cells. NK cells serve as the first line of immune defense against cancer cells, however, increasing evidence shows emergence of dysfunctional phenotypes. In this study, we performed a thorough characterization, and uncovered the underlying etiologies of the "anergic" and "exhausted" dysfunctional states. Although many surface markers have been suggested (Judge et al, 2020), the

molecular and transcriptional wiring remains unexplored. We demonstrate that the naturally existing "anergic" NK cell population shares phenotypic, transcriptomic, and functional similarity with the canonical NK cell "exhaustion" state arising from the TME. DGKα and Egr2 were identified as key intrinsic regulators governing NK cell dysfunctional states of "anergy" and "exhaustion". Our findings indicate that Egr2 serves as the primary transcription factor accounting for the intrinsic non-functionality of these cells. Moreover, our transcriptome analysis reveals sharing of key genes between anergic/exhausted NK cells, and exhausted CD8$^+$ T cells (Fig. 2D–G), including *NFAT2, EGR2, PDCD1*, and *TOX2*, revealing transcriptional commonalities. In this regard, targeting a shared intrinsic regulator such as Egr2, identified here, for therapy can improve the response of the TILs as a whole rather than only a specific immune cell population.

The dysregulation of NK cell function is a significant hurdle in immunotherapeutic approaches, such as chimeric antigen receptors (CAR) and immune checkpoint blockade (ICB) including anti-CTLA-4 and anti-PD-1 treatments despite the great interest in NK cells as a candidate for immunotherapy (Ben-Shmuel et al, 2020; Laskowski et al, 2022). While these therapies have shown promise, cell evasion and exhaustion remain persistent challenges, limiting efficacy, especially of CAR– T and NK cells (Valeri et al, 2022; Kouro et al, 2022; Titov et al, 2022; Selli et al, 2023; Good et al, 2021). Adoptive transfer of NK cells, a common approach, also leads to reduced cytotoxic potential due to loss of NK activity during ex vivo expansion of NK cells (Judge et al, 2020; Gill et al, 2012). ICB therapies target one or more checkpoints such as the surface markers PD-1, CTLA4, TIGIT, or NKG2A (Leach et al, 1996; Ott et al, 2013; Pan et al, 2023). Targeting these suppresses only some of the pathways that are associated with the particular checkpoint inhibitor used, and these checkpoint blockade strategies only extend the cytotoxic window (Barber et al, 2006), but do not reverse the exhaustion-associated transcriptional imprint (Pauken et al, 2016).

Along these lines, our study highlights two key intrinsic checkpoints in NK cell dysfunction (Fig. 5J); (i) The TF Egr2 establishes a molecular circuitry based on DGKα transcription leading to NK cell anergy. (ii) DGKα facilitates conversion of DAG to PA, leading to impaired signaling cascades through the recruitment of the phosphatase SHP-1 to the cell membrane, resulting in dephosphorylation of its targets, e.g., PLCγ1/2 (Fig. EV4C–E), LAT, and ZAP70, suppressing the signaling cascade, and resulting in the dysfunctional phenotype. Thus, the efficacy of EGR2 silencing observed in our results occurs through reduced DGKα expression limiting SHP-1 recruitment by PA, thereby enabling PLCγ1/2 enzymatic activity (Matalon et al, 2016; Bradshaw and Dennis, 2010; Frank et al, 1999). This facilitates the initiation of the DAG-activated PKCθ-IKK-NFkB axis and RasGRP-MAPK pathway (Chen et al, 2016b; Krishna and Zhong, 2013) together with IP$_3$-mediated calcium-calcineurin-NFAT signaling cascades (Vig and Kinet, 2009). In addition, EGR2 gene silencing shifted the dysfunctional phenotype from a PD-1$^{high}$ to PD-1$^{low}$ profile (Fig. 4G), implying that the TF Egr2 induces PD-1 expression. Notably, the potent recovery of overall functionality and molecular wiring observed following EGR2 silencing, compared to the weaker effect of DGKα, ζ silencing, reinforces the crucial role of Egr2 as a central regulator and a potent therapeutic target to restore NK cell functions.

The involvement of SHP-1 in NK cell signaling has been intensively studied by us (Matalon et al, 2016, 2018; Ben-Shmuel et al, 2022; Biber et al, 2021; Ben-Shmuel et al, 2021) and others (Wu et al, 2021; Schmied et al, 2023), yet the mechanism underlying its transcriptional regulation in anergy remains unclear. Our data implies that EGR2 gene silencing downregulates SHP-1 activity, potentially via enhancing DAG availability by reducing DGKα levels. This, in turn, initiates the activation of PKCθ, which modulates SHP-1 activity through serine 591, supported by our recent data (Ben-Shmuel et al, 2022).

Furthermore, the RNA sequencing analysis and the in vivo model shown here suggest that Egr2 potentially cooperates with other TFs, presumably NFAT2, to initiate the dysfunction-based transcriptional program along with high mobility group (HMG)—box TFs such as TOX2, and members of the NR4a family, which are validated targets of calcium-calcineurin regulated NFAT2 in the absence of partners such as AP-1 and NFkB (Wagle et al, 2021; Seo et al, 2019; Sekine et al, 2020). The involvement of Egr2 and potentially NFAT2/ TOX2/other TFs in the terminal regulation of anergy highlights the complexity of the transcriptional program associated with dysfunctional NK cells. This finding establishes a functional link between the naturally induced "anergy" and the "exhausted" state of NK cells within the TME. Thus, targeting the intrinsic regulators of dysfunctional NK cells (i.e., Egr2), leads to the rewiring of NK cell functional circuitry, overcomes the dysfunction-associated transcriptional imprint, and potentially enhances the efficiency of immunotherapies.

On this note, depletion of dysfunctional cells from the TME was suggested to improve therapeutic efficacy. Our study provides a novel strategy for in situ reprogramming of these dysfunctional cells rather than depleting them (Marcucci and Rumio, 2021; Cook and Whitmire, 2013). We describe a potential platform for this approach using an NP drug delivery system (Biber et al, 2021) (Fig. 5C). In addition, our in vitro 3D OTS CLL model and in vivo PDAC model of NK cell exhaustion demonstrate that EGR2 silencing, not only restores the functionality of anergic cells but also enables these NK cells to potentially control solid tumor growth and influence disease outcomes. This potentially serves as proof of concept for the possibilities of reprogramming dysfunctional NK cells in situ, bypassing the need for adoptive transfer.

While the majority of studies on anergic NK cells have focused on their presence in peripheral blood, evidence suggests that NK cells develop and mature in different tissues including spleen, liver, bone marrow, lymph node, and lung (Dogra et al, 2020; Hashemi and Malarkannan, 2020). This leads to the notion that the NK cells undergo education in the respective niche, resulting in the potential emergence of anergic phenotype. The characterization and the functional implications of anergic NK cells arising in different tissue contexts are still unknown. Further research is necessary to comprehensively describe the presence and functional characteristics of these potential population(s). Furthermore, the complex interplay between anergic cells and the TME requires in-depth investigation to unravel underlying mechanisms and their potential implications. In this regard, our NP-based approach to reprogramming these dysfunctional cells in their milieu holds great promise. Reversing both NK cell "anergy" and NK cell "exhaustion" is essential as it restores a crucial component of the immune system's capacity to detect and eliminate cancer cells, potentially leading to more effective immunotherapeutic strategies and better patient outcomes.

In summary, our study not only elucidates the molecular framework of NK cell anergy but also highlights its functional, phenotypical, and transcriptional parallels with canonical NK cell exhaustion. In addition, we identified shared intrinsic factors that can be targeted and reversed through immunotherapy. Furthermore, our results provide valuable insights into the mechanisms and signaling pathways responsible for NK cell dysfunctional states. Focusing on the regulation of NK cell anergy and exhaustion will potentially lead to the development of strategies to modulate the anergic state in settings of autoimmunity and transplantation or to enhance NK cell function in cancer and chronic viral infection.

# Methods

## Cells

Human B-cell lymphoma 721.221 cell lines (RRID: CVCL_6263), and K562 (ATCC- CCL-243) CML cell lines were generously provided by Prof. Ofer Mandelboim (Department of Microbiology and Immunology, Faculty of Medicine, Hebrew University, Israel). 721.221 HLA-negative and 721.221 expressing HLA-Cw7 cells were cultured in RPMI 1640 (Sigma-Aldrich) supplemented with 10% FBS, 2 mM L-glutamine, penicillin (50 µg/ml), streptomycin (50 µg/ml), 1% non-essential amino acids, and 1% sodium pyruvate. Stable 721.221 HLA Cw7 cell lines expressing mCherry were previously established in our laboratory (Ben-Shmuel et al, 2022; Biber et al, 2021) and were used for Incucyte assay. The cells were maintained in culture at a density of 0.1–0.25 × 10⁶ cells/mL. The human pancreatic ductal carcinoma, PANC-1 cell line was used (ATCC – CRL1469) for the in vivo tumor experiments. For the in vivo experiments, the PANC-1 cells used were sourced from passage 4–8 which were maintained at a confluence of 60–80%. Human K562 CML cell lines stably expressing CFP were used for the OTS system (Ben-Shmuel et al, 2022). The cells were maintained in culture at a density of 0.3–0.5 × 10⁶ cells/mL.

## PBMC isolation

Human primary PBMCs were isolated from the whole blood of healthy donors or patients, as previously described (Barda-Saad et al, 2005; Ben-Shmuel et al, 2022; Biber et al, 2021). Blood samples from healthy donors were randomly collected and provided by Magen David Adom (MDA) (Israeli National Blood Bank, no. 12-0016). Informed consent was obtained from all donors. The donor's identification information remained anonymous. The research was performed with the approval of and according to the guidelines of the Bar-Ilan University Ethics Committee. Briefly, human PBMCs were isolated from whole blood by Ficoll-Histopaque density (1.077 g/mL) gradient centrifugation (MP Biomedical). Isolated cells were cultured in RPMI 1640 (Sigma-Aldrich) supplemented with 10% FBS (Biological Industries). All cells were maintained at 37 °C containing 5% CO₂.

## Primary NK cell isolation and sorting

Primary NK (pNK) cells were isolated from PBMCS of healthy donors using the EasySep™ human NK cell enrichment kit (STEMCELL Technologies) with a purity of CD3⁻CD56⁺ NK cells >95%. The efficiency of the magnetic separation was checked after each separation via flow cytometry. Subsequently, pNK cells were stained with pan anti KIR2D anti KIR3DL1 and anti-NKG2A antibodies (Miltenyi biotec and BioLegend), and then subjected to cell sorting using Aria Cell Sorter II (Fig. EV1A). PBMCs were used as a negative control during sorting. NK cell isolation and sorting was 95% accurate, and the resulting cells were then used for functional assays.

Sorting was conducted for pNK obtained from a healthy individual donor in every experiment involving anergic and responsive NK cells.

## Antibodies

Antibodies and their sources were as follows.

Primary antibodies for immunoprecipitation and immunoblotting: rabbit anti-Cbl, mouse anti-GAPDH (sc-0411), rabbit anti-GAPDH (FL-335), mouse anti-pERK Tyr204 (Santa Cruz Biotechnology); rabbit anti-Egr2 (Sigma Aldrich), rabbit anti-DGKα (Proteintech); mouse anti-NFAT1 (Abcam), mouse anti- NFAT2 (Abcam), rabbit anti-DGKζ (Abcam); rabbit anti-pPLCγ1(Y783), rabbit anti-pSHP-1 S591(ECM Biosciences).

Primary antibodies for FACS: Surface staining (FACS): PE–anti-Human KIR2D, PE–anti-Human NKG2A (Miltenyi Biotech); PE–anti-Human KIR3DL1, Alexa 647-anti-Human Tim-3, Alexa 647-anti- Human PD-1, Alexa 647–anti-Human TIGIT, FITC anti-Human CD107a, mouse anti- Human CD16, FITC-anti-Human CD56 (BioLegend); mouse anti-Human NKp46 (Merck Millipore), FITC-anti Human CD45, PE-Cy5 anti-Human CD3 (BD Biosciences).

Secondary antibodies: goat anti-mouse IgG (Jackson ImmunoResearch), goat anti-rabbit IgG (Jackson ImmunoResearch), goat-anti-mouse Alexa 647 (A21235) (Invitrogen).

More information on all the antibodies and reagents used in the study is available in Appendix Table S1.

## Cell transfection by electroporation

pNK cells were transfected with a Lonza Nucleofector™ 2b device using the manufacturer's protocol U-001, with either non-specific (NS) siRNA, DGKα, ζ siRNA or Egr2 siRNA. Transfections were performed 48 h before biochemical and functional assays.

## Lysis and western blot of NK cells

Western blotting was performed as previously described (Ben-Shmuel et al, 2022).

## Cytotoxicity assay/ [³⁵S] Met release assay

Cytotoxicity assays were performed as described (Ben-Shmuel et al, 2022; Biber et al, 2021).

## Degranulation assay

The degranulation assay based on staining for CD107a was performed as described (Ben-Shmuel et al, 2022; Biber et al, 2021).

## Flow cytometric analysis

**(i) Surface receptor staining:** $5 \times 10^5$ pNK cells were incubated for 30 min on ice with 1:25 diluted anti-PD-1- Alexa 647 or anti-Tim-3-Alexa 647 or anti-TIGIT-Alexa 647-conjugated antibodies. The cells were then washed twice with Phosphate Buffer Saline (PBS) and analyzed by flow cytometry. pNK cells were stained anti-NKG2A, pan KIR2D, KIR3DL1, CD3, CD56, PD1, TIGIT, TIM-3, CD16, NKp46 for 10 min in 4 °C or 30 min on ice according to the manufacturer's protocols, and then acquired using the ARIAIII or Fortessa FACS, and analyzed by Flow Jo Software v10.8.1.

(ii) **Intracellular staining:** NK cells (anergic and responsive) ($0.5–1 \times 10^6$ cells per sample) and 721.221 target cells were incubated separately on ice for 10 min, at a ratio of 1:2. The cells were mixed, centrifuged, and incubated on ice for 15 min. The cell mixture was then transferred to 37 °C for the indicated period of time, and subsequently fixed with 0.4 ml 3.7% paraformaldehyde for 20 min at room temperature (RT). After two additional washes, cells were permeabilized with 0.1% Triton X-100 in PBS for 4 min at RT. Cells were then blocked for 45 min with 2% goat serum, followed by 1 h incubation with relevant antibody [pERK (Y204) (on ice)]. The samples were washed twice, and then incubated with 1:2500-diluted Alexa Fluor 647-conjugated goat anti-Rabbit IgG antibody (Invitrogen), for 30 min on ice. Fluorescence was analyzed by flow cytometry on a BD Fortessa system using FACSDiva Software. For all experiments involving FACS, the gating strategy is shown in the extended view figures.

## Measurement of intracellular calcium ($Ca^{2+}$) concentration

Intracellular calcium was determined as described using Indo-1 AM staining (Sabag et al, 2022; Ben-Shmuel et al, 2022; Biber et al, 2021).

## Incucyte killing Assay

Anergic and responsive cells were sorted as described above. 721.221 HLA-Cw7 cells ($1 \times 10^4$) expressing mCherry were seeded in 100 μL Opti-MEM in Corning 96-well flat bottom plates (Cat no. TCP011096). Effector cells, consisting of anergic or responsive cells, treated with Egr2 siRNA or N.S. siRNA were seeded along with Cw7 cherry cells at an E:T ratio of 10:1. Cw7 cherry cells alone were used as target-only control for normalization of fluorescence values. The plates were maintained in the Incucyte SX5—Live-cell analysis and analyzed using Incucyte Spectral analysis software—basic analyzer for 14 h. The phase confluences were divided to orange confluence, and mean intensity was calculated and normalized to time point 0:00 h. The graph represents the decrease in fluorescence (*Y* axis) plotted against time points in hours (*X* axis). Representative images with respective normalized fluorescence values are depicted in the Fig. EV5. The analysis was performed based on 9 individual fields per image for 16 images per experimental condition, each having approximately $N = 50$ cells per image.

## Liposome nanoparticle (NP) preparation

NP-based siRNA delivery particles were prepared as previously described by us (Biber et al, 2021). The NPs were tagged with anti-NKp46 and labeled with Rhodamine to enable detection. Egr2 siRNA or NS siRNA were encapsulated by protamine entrapment as previously established (Biber et al, 2021).

## OTS killing assay—3D OTS model of CML

K562 CFP ($1*10^5$) (using stable lines previously established in the lab (Ben-Shmuel et al, 2022)) were mixed with Matrigel (Phenol Red free. Cat no. #) at a 1:1 ratio (v/v—25 μL:25 μL) in Corning 24-well flat bottom plates. The cells were cultured using OTS media, RPMI media supplemented with 0.1 μg/mL fibroblast growth factor (FGF), 0.18 μg/mL epidermal growth factor (EGF) and 20 ng/mL transforming growth factor (TGFβ) and allowed to form domes for 48 h. pNK cells were obtained from PBMCs of healthy donors and subjected to sorting to obtain anergic and responsive populations. They were then seeded on the OTS at an E:T ratio of 5:1 in Opti-MEM for imaging. Egr2 siRNA or NS siRNA were encapsulated in anti-NKp46-tagged liposomal nanoparticles using a previously established protocol in our lab (Biber et al, 2021) and delivered to the OTS post 6 h. Cultures were then subjected to the incucyte killing assay by measuring the decrease in fluorescence from the target cells (CFP) for 48 h; K562 CFP cells alone were taken as OTS-only control for normalization of values. The images were obtained using Standard Analyzer with a 20× objective. Whole well images were obtained using 4X objective at 48 h. The plates were maintained in the Sartorius SX5 Incucyte and analyzed using Incucyte Spectral analysis software, basic analyzer and adherent cell-by-cell analysis, for 48 h. The phase confluences were divided to green confluence, and mean intensity was calculated and normalized to the measurement at the initiation of the experiment. The graph represents the decrease in fluorescence (*Y* axis) plotted against time in hours (*X* axis).

## RNA interference

siRNAs specific for human EGR2, DGKα, DGKζ and non-specific controls were purchased from Sigma-Aldrich. For the knockdown of Egr2 and DGK proteins, specific oligonucleotides encoding siRNAs were as follows:

esiEgr2 MISSION Sigma-Aldrich Cat no. EHU124311, DGKα siRNA Sigma-Aldrich Cat no. SASI_HSS01_00072301, DGKζ siRNA Sigma-Aldrich Cat no. SASI_HS02_00324291.

Non-targeting (non-specific), negative control siRNA duplex: sense 5′-UAGCGACUAAACACAUCAA-3′, anti-sense 5′-UUGAUGUGUUUAGUCGCUA-3′.

## RNA isolation

Primary NK cells after sorting were processed for RNA isolation. Isolation was performed based on the Quick-RNA Microprep (Cat no. R1050 & R1051; ZymoResearch) according to the manufacturer's protocol.

### RNA Sequencing and library preparation

The libraries were sequenced in multiplex as single-end, non-strand specific 81-bp reads on an Illumina NextSeq 550 at the Bar-Ilan University Sequencing Unit, resulting in an average of 30–50 million reads per sample. The RNA sequencing data was trimmed using Trim galore (version 0.5.0) and aligned using STAR (v2.7.3a)

(Dobin et al, 2013) with default parameters. The ENSEMBL human hg38 build and its corresponding GTF were used to build the STAR reference database. The number of reads per gene was quantified using HTSeq-Count (v0.6.1p1) (Putri et al, 2022; Anders et al, 2015). The R (v4.1.1) Bioconductor package, DESeq2 (v1.32.0) (Love et al, 2014) was used to classify genes as differentially expressed (Benjamini-Hochberg adjusted $P$ value < 0.05).

## Xenogeneic tumor-grafted mouse models

NOD/SCID IL-2Rγ$^{null}$ (NRG) mice grafted with aggressive, NK-resistant pancreatic cancer cells (PANC-1 – ATCC CRL1469) were used to examine NK cell dysfunction. All experiments were performed in accordance with Bar-Ilan University Ethics rules, under protocols approved by Bar-Ilan University. Mice were maintained under specific-pathogen-free conditions, with a 12-h night/daylight cycle, and at stable ambient temperature with 40–70% relative humidity. Mice were monitored daily and were euthanized upon manifestation of hunched posture, impaired mobility, rough coat, or paralysis. Details on the in vivo conditions and study designs have been included in Table 1. In all the in vivo experiments, a total of three mice were euthanized prior to the endpoint of the experiment and were not included in the results.

For establishing the xenograft model, 6-week-old female NRG mice (Jackson Laboratories) were infused *s.c.* with $4 \times 10^6$ PANC-1 cells in 150 μL of PBS, and tumors were allowed to reach sizes of ~250–300 mm³. Mouse weight, signs of stress, and tumor size were monitored daily. Tumor volume was measured using a digital caliper and calculated using the following formula: Volume (mm³) = 0.5*(major axis)*(minor axis)² (Tomayko and Reynolds, 1989). The average tumor growth rate was calculated according to the formula: Average growth rate = (Current tumor size – Initial tumor size)/Elapsed treatment time (days) (Tomayko and Reynolds, 1989). Animals were randomized to ensure uniform tumor burden before primary NK cell treatment. Human pNK cells isolated from the peripheral blood of healthy donors were isolated and suspended in 150 μL PBS containing $1.1 \times 10^7$ pNKs and injected *i.t.* The tumors were excised 4 days after the pNK injection, treated with collagenase IV and DNAse in PBS$^{-/-}$ and incubated for 45 min at 37 °C with continuous orbital shaking at 150 rpm. The end point was chosen based on tumor reduction plateau, suggesting the inability of the NK cells to control the tumor growth rate. The tumors were then strained using size-specific strainers, washed twice and dissociated into single-cell suspensions followed by two washes in RPMI 1640. The resulting cell suspension was used for functional assays.

For the in vivo experiment showing the ability of Egr2-silenced pNK to control tumor growth, PDAC xenograft mice were established as described above. At day 9, 3 days following pNK injection ($1.1 \times 10^7$ cells), 300 μg NPs encapsulating EGR2 or NS siRNA suspended in 150uL PBS were infused *i.v* and re-administered every 3 days. Tumor size and growth rates were monitored on a daily basis using caliper measurements. Randomization was done to ensure equal tumor burden before human pNK cell treatment. Tumor size and growth rate were calculated throughout the experiment. Mice were inspected daily for general well-being, and body score index (BSI); at the first indication of morbidity (weight loss, lethargy, ruffled fur) or 48 h following the last NP injection, mice were euthanized by $CO_2$, and tumors were harvested (as mentioned above) for ex vivo analysis.

## Sequencing data processing and gene-set comparisons

PCA analysis was performed using DESeq2 on the variance-stabilized transformed (vst) normalized RNA-seq data. The heatmap was generated using the R package pheatmap (v1.0.12) (Raivo, 2018). The volcano plot was generated using GraphPad v9.0.1, and line graphs was generated using the R package ggplot2 (v 3.3.5) (Wickham, 2016) and GraphPad v9.0.1. The java-based

**Table 1. In vivo conditions and study design.**

| In vivo conditions and study design | |
|---|---|
| Mouse model | Female SCID-NRG mice |
| Cell type and model | PANC-1 cells lines—pancreatic ductal adenocarcinoma |
| Number of positions injected within each mouse | 1 |
| Number of mice per cage | Five mice per cage |
| Immune cells used in the study | Primary human NK cells isolated from healthy donors and checked for purity—CD3⁻CD56⁺ before administration |
| Treatment: intratumor administration of PANC-1 | $4 \times 10^6$ cells in 100 μL PBS |
| Treatment injection interval of human pNK | Every 3 days for six injections—$1.1 \times 10^7$ cells—intratumor (*i.t*) |
| Administration of nanoparticles | 300 μg NPs every 3 days, intravenous (*i.v*) |
| Homing | SCID/NOD NRG female, 6–8-week-old mice were purchased from Envigo or Jackson laboratories. All mice were housed in *IVC* caging, supplied with irradiated shredded corn cob bedding and irradiated mouse feed diet. The light–dark cycle was 12 h. The ambient temperature of each room was set at 20 °C ± 1 °C. The temperature inside the boxes generally remained between 22 °C and 24 °C. Humidity was set between 35% and 55%. |
| Ethics oversight | Housing and breeding of mice and experimental procedures were performed according to the guidelines of the Bar-Ilan University Animal Ethics Committee (#82-10-18). |
| Study design | Randomization was used to divide the animals for in vivo treatments. |

GSEA Desktop Application v4.2.2 (Subramanian et al, 2005) was used to perform GSEA of gene sets from viral T-cell exhaustion datasets (West et al) and T-cell exhaustion datasets (Scott et al, Martinez et al). The references for the software are included in the appendix file.

## Analysis of cancer patient data

Survival Genie open-source software was used to perform survival analysis on AML and LGG datasets obtained from TCGA (Dwivedi et al, 2022). The gene set was defined according to the NK cell signature established by Böttcher et al—*NCR1, NCR3, KLRB1, CD160*, and *PRF1* (Böttcher et al, 2018) along with the genes—*DGKA-EGR2* low and high quartiles.

## Statistical analysis

The data and graphs were processed using Microsoft Excel (v14.0.7) and GraphPad Prism9, and graphical results are reported throughout as mean ± SEM. '*n*' represents the number of independent experiments, and the number of healthy donors used to obtain the data, unless mentioned otherwise. The means of the two sample groups were compared using an unpaired two-tailed Student's *t* test. The interaction value between different treatment groups (control vs experimental group) were analyzed by one-way ANOVA and Tukey's post hoc multiple comparison, wherever required after normalization to the control group. For all one-way ANOVA and Student's *t* test analyses, we confirmed equality of variance and normality using Shapiro–Wilk and Kolmogorov–Smirnoff tests. For experiments including >2 treatment conditions, two-way ANOVA and Tukey's post hoc multiple comparison were used. For the GSEA analysis, the weighted Kolmogorov–Smirnov statistical test was used. A *P* value less than 0.05 was regarded as statistically significant. All *P* values are indicated within the graph along with the independent experimental repeats.

## Data availability

All data attained to support our conclusions described in this manuscript are presented in the paper and its supplementary materials (Appendix Fig. S1, source data files). RNA-Seq data: Gene Expression Omnibus GSE248228.

## Peer review information

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

## Acknowledgements

The authors thank Tammir Jubany, Hadar Krap Shachar, Stephanie Wassermann, Matan Avivi, Dr. Hagit Hauschner and Dr. Irit Shoval, from Bar-Ilan University, and Drs. Liron Miller and Natalie Landa from the blood bank and transfusion center of Sheba Medical Center, Israel, for their technical assistance. The project was partially funded by the Israel Science Foundation (ISF), 1001/23 and part by the Ministry of Innovation, Science and Technology no. 0005925.

## Author contributions

**Batel Sabag**: Conceptualization; Data curation; Formal analysis; Validation; Investigation; Visualization; Methodology; Writing—original draft. **Abhishek Puthenveetil**: Conceptualization; Data curation; Software; Formal analysis; Validation; Investigation; Visualization; Methodology; Writing—original draft. **Moria Levy**: Data curation; Methodology. **Noah Joseph**: Methodology. **Tirtza Doniger**: Software; Formal analysis; Methodology. **Orly Yaron**: Methodology. **Sarit Karako-Lampert**: Methodology. **Itay Lazar**: Resources; Supervision. **Fatima Awwad**: Data curation; Methodology. **Shahar Ashkenazi**: Methodology. **Mira Barda-Saad**: Conceptualization; Resources; Supervision; Funding acquisition; Validation; Investigation; Writing—original draft; Project administration; Writing—review and editing.

## Disclosure and competing interests statement

The authors declare no competing interests.

# Expanded View Figures

**Figure EV1.   RNA seq analysis—gene and pathway enrichment.**

(**A**) Graphical presentation of NK isolation process -gating strategy. Primary human NK cells were isolated from PBMCs using negative selection; following purity check by CD3⁻CD56⁺ expression, the cells were stained for NKG2A PE and panKIR PE and further sorted to anergic and responsive NK cell subsets based on PE expression (NKG2A⁻ panKIR⁻ (PE negative) anergic subset; NKG2A⁺ panKIR⁺ (PE positive) responsive subset). (**B, C**) Purified anergic and responsive cells were subjected to incubation with 721.221 No HLA cell lines (E:T - 1:3) for 5 h at 37 °C, and were subjected to degranulation assay ($n = 3$ healthy donors). $P$ values were calculated using two-tailed paired $t$ test and are represented within the graph presented as means ± SEM. (**B**) For the S³⁵ assay the target cells were labeled with S³⁵, and tumor lysis was measured ($n = 3$, where n is the number of healthy donors used to obtain the pNK cells). $P$ values were calculated using a two-tailed paired $t$ test and are represented within the graph presented as means ± SEM. (**C**) Metascape analysis of key genes and the associated enriched pathways from KEGG and GO, upregulated in the (**D**) responsive, and (**E**) anergic populations ($n = 4$, where n is the number of healthy donors used to obtain the pNK cells). (**F, G**) Metascape analysis of key genes and the enriched terms annotated to pathways from various datasets associated with the top-hit significant genes (**F**), and the TFs reflecting the top-significant genes upregulated in the anergic subset, obtained via enrichment analysis in TRRUST (**G**). $P$ values for (**D–G**) were obtained using Hypergeometric test and Fisher's exact test. (**H**) The interactome cluster of the enriched terms annotated to pathways obtained from the significant DEGs. Source data are available online for this figure.

                                        

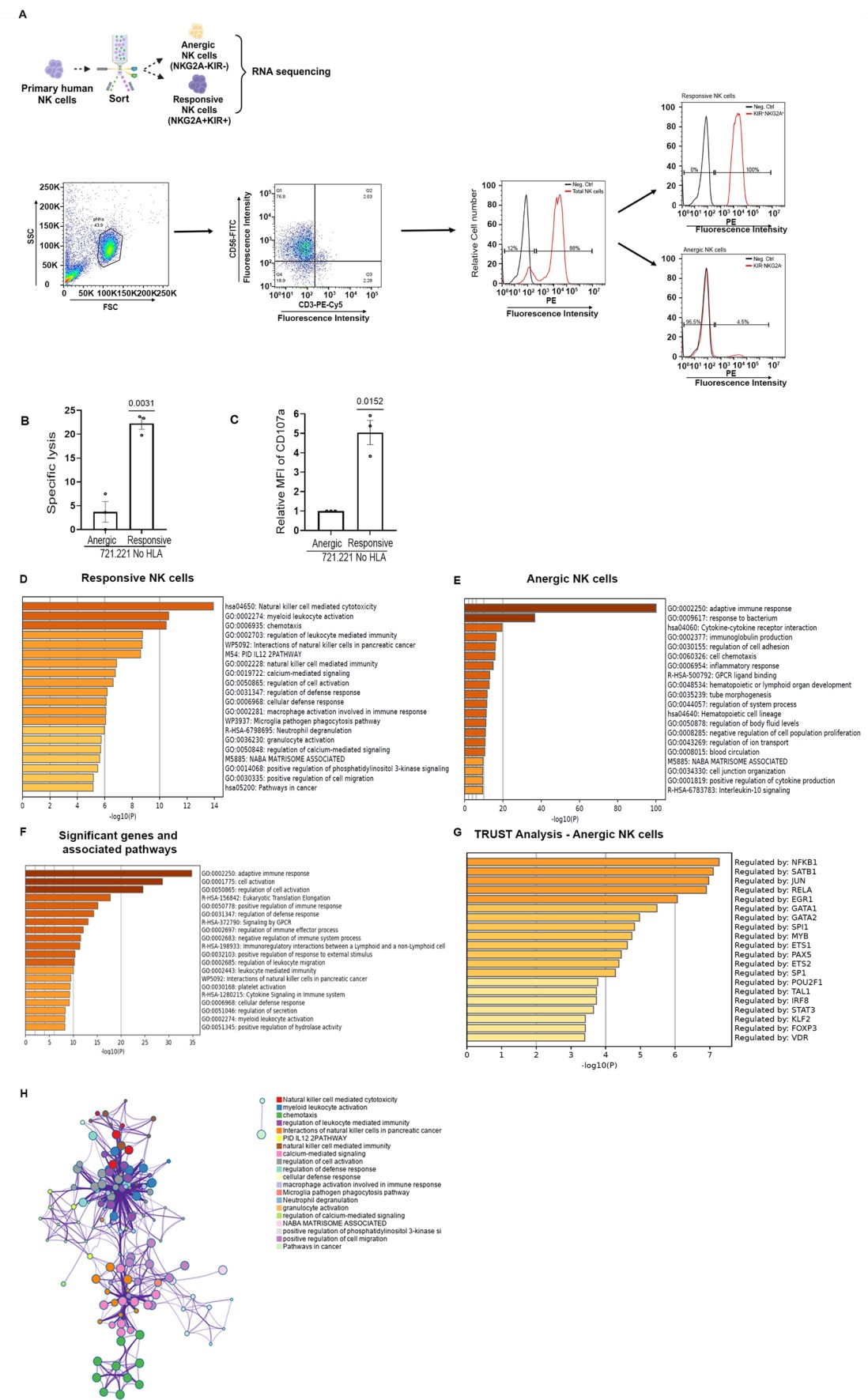

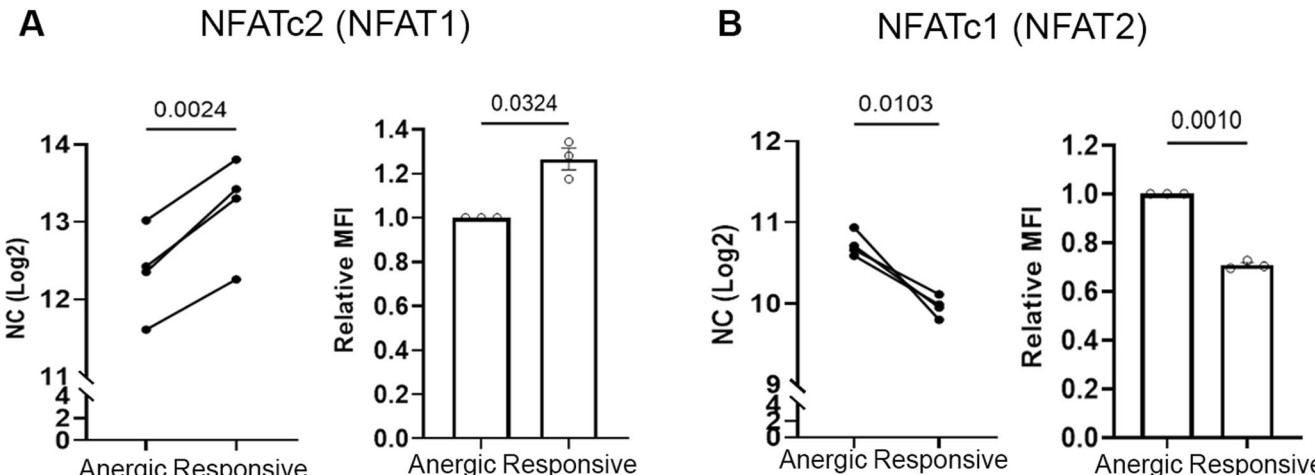

**A** NFATc2 (NFAT1)

**B** NFATc1 (NFAT2)

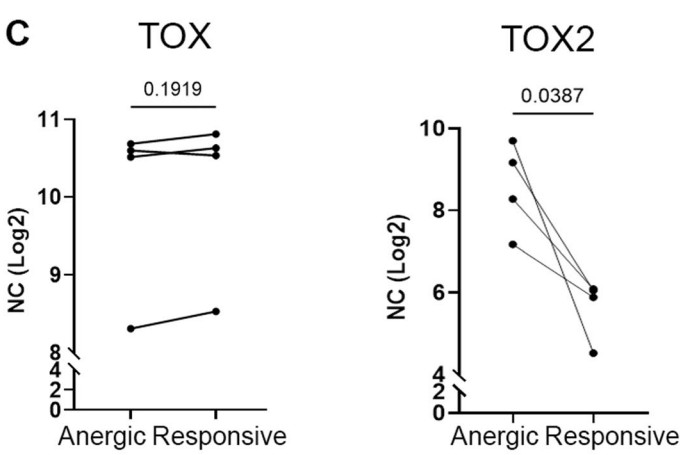

**C** TOX  TOX2

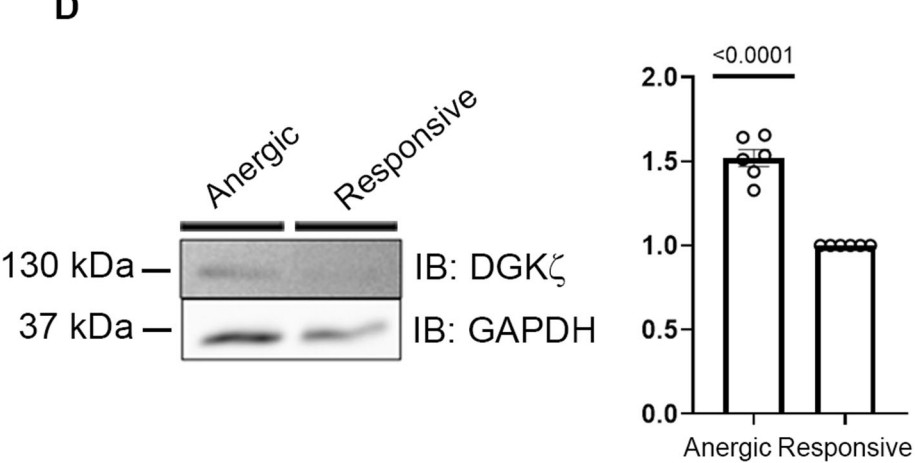

**D**

◄ **Figure EV2.  RNA seq analysis and DGKα expression levels.**

(A, B) Transcript level by RNA seq (left) ($n = 4$, healthy donors used to obtain the pNK) and protein expression ($n = 3$ healthy donors) via flow cytometry (right) of (A) NFAT1 (NFATc2) and (B) NFAT2 (NFAT c1), respectively. The *P* value was calculated using a two-tailed *t* test, and the number of repeats is indicated within the graph presented as means ± SEM. (C) The transcript levels of TOX and TOX2 in anergic vs responsive cells ($n = 4$, where *n* is the number of healthy donors used to obtain the pNK cells). *P* values were calculated using a two-tailed *t* test with pairing and are indicated within the graph; NC normalized counts. (D) Purified anergic and responsive NK cells were lysed and subjected to western blot analysis with DGKα antibody ($n = 6$, where *n* is the number of healthy donors used to obtain the pNK cells). *P* values were calculated using a two-tailed *t* test with pairing, and are indicated within the graph presented as means ± SEM. Source data are available online for this figure.

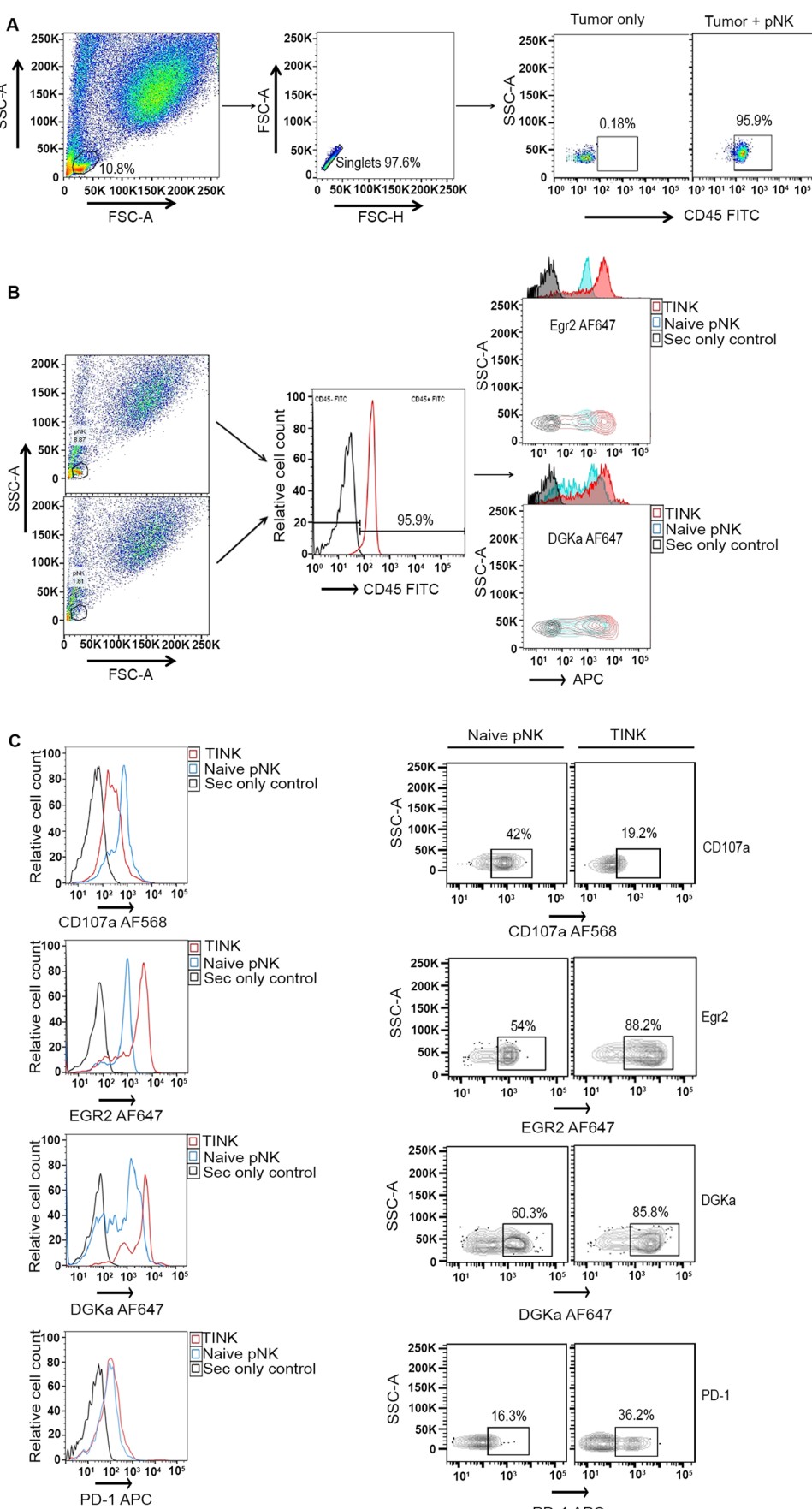

**Figure EV3. Flow cytometry gating strategy.**

The gating strategy employed for the in vivo experiment. (**A**) NK cells were distinguished from the target PANC-1 cells according to FSC and SSC. CD45 (antibody specifically recognizing human (h)CD45) expression was used to further distinguish the pNK from the tumor and any other murine cells. Pseudocolor presentation is shown to identify the NK cells, which are CD45$^+$ (95.9%) referred as tumor infiltrating NK cells (TINK). (**B**) Flow cytometry analysis of intracellular staining was performed to measure pNK expression on cells from the mice with tumor-only (no NK administration) control and distinguished based on CD45 expression. The NK cells were distinguished from the target cells based on their volume and density (FSC and SSC) and were re-gated to CD45$^+$ subsets. DGKα and Egr2 expressions were measured on the CD45$^+$ gated population. Histogram offsets and MFI were used for graphical presentation. Graphs show fluorescence intensity. The *Y* axis indicates relative cell number, and the *X* axis indicates the MFI. (**C**) Representative histograms and contour plots including outliers for the data in Fig. 3C–F.

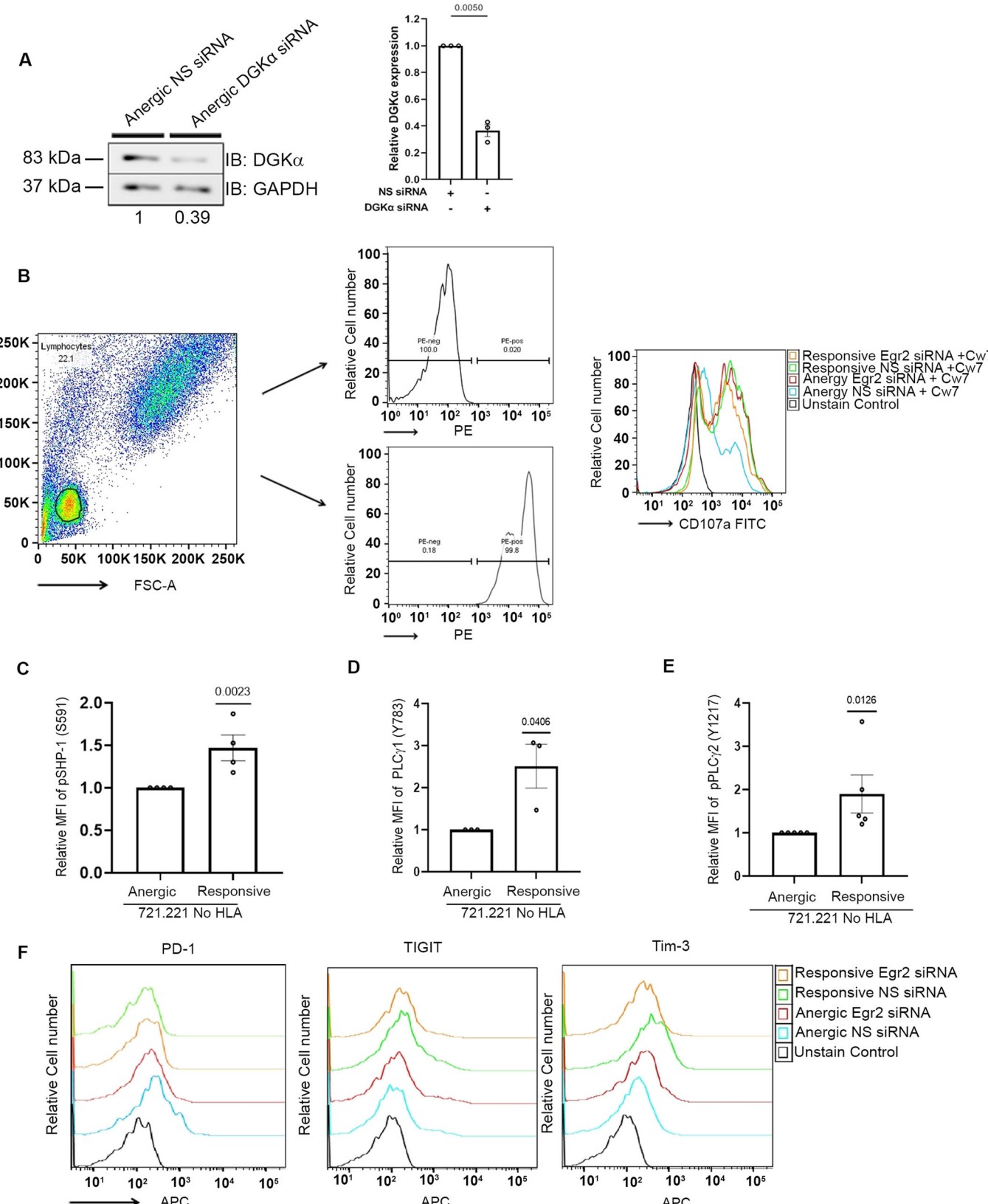

◀  **Figure EV4.   Silencing efficiency and flow cytometry gating strategies.**

(**A**) Gene silencing of DGKα. Anergic cells were treated with DGKα siRNA or NS siRNA and were then lysed and subjected to western blot analysis. One blot representative of three experiments is shown. The right panel shows a graph representing the quantified blots ($n = 3$, where n is the number of healthy donors used to obtain the pNK cells). Data are presented as mean ± SEM. *P* value was calculated using a two-tailed paired *t* test and is indicated within the graph. (**B**) Gating strategy employed for the analysis of results in Fig. 4. The NK cells and the 221 HLA-Cw7 cells were differentiated based on size and granularity (FSC-SSC) and gated for PE to distinguish the anergic (PE$^-$) versus the responsive population (PE$^+$). They were subsequently gated for CD107a, as indicated on the overlaid histograms. (**C–E**) Purified responsive and anergic NK cells were stimulated with 721.221 target cells, lysed and subjected to FACS analysis with (**C**) anti-pSHP-1(S591), (**D**) anti-pPLCγ1 (Y783), and (**E**) anti-pPLCγ2 (Y1217) antibodies. Graph summarizing the MFI of pSHP-1 ($n = 5$), pPLCγ1 ($n = 3$), and pPLCγ2 ($n = 5$) expression levels (where *n* is the number of healthy donors used to obtain the pNK cells). Data are presented as mean ± SEM. *P* value was calculated using a two-tailed *t* test with matched data repeats and is indicated within the graph. (**F**) Representative overlaid histograms for Fig. 4G–I. Source data are available online for this figure.

**A**

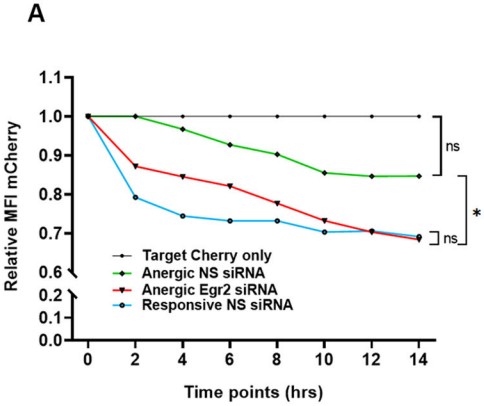

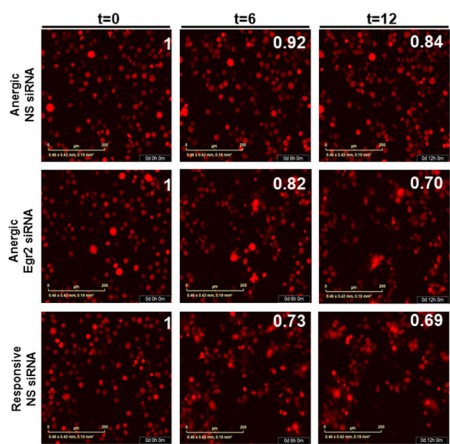

**B**

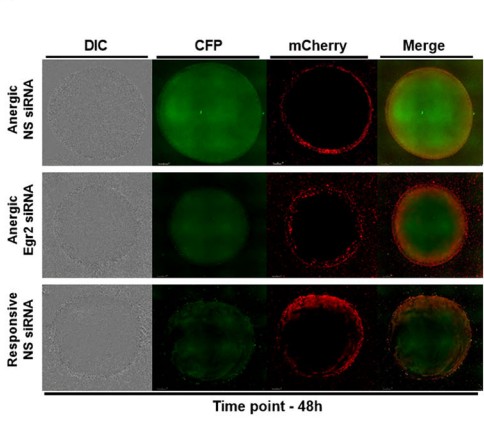

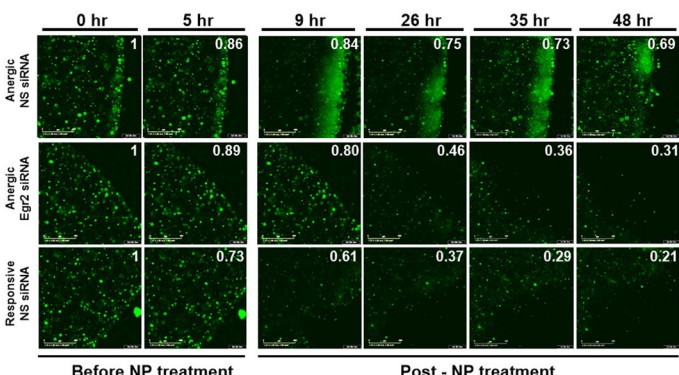

**C**

**D**

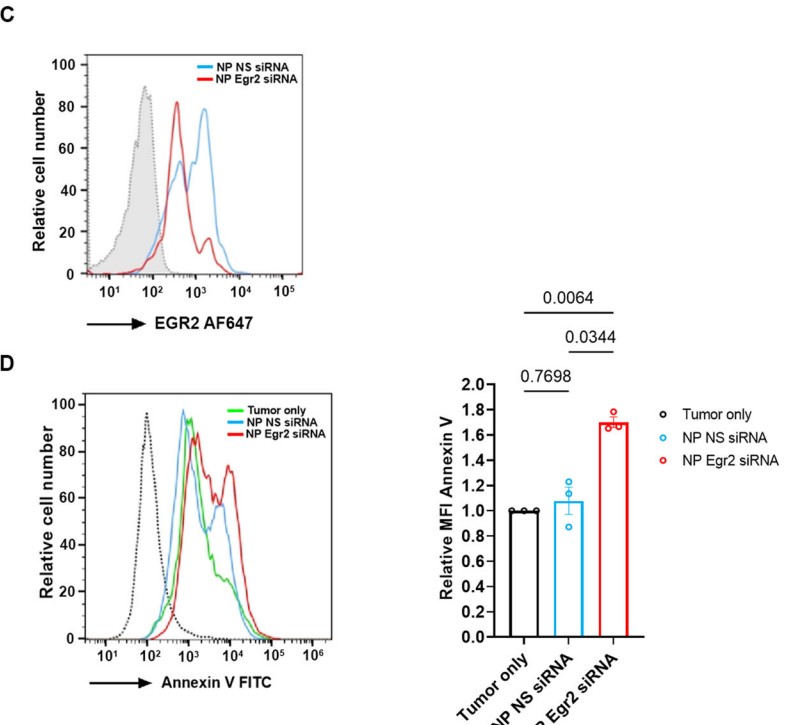

Figure EV5.   **Incucyte analysis with representative images and ex vivo analysis.**

(**A**) Anergic and responsive cells were treated with either Egr2 or NS siRNA, and Incucyte-based tumor lysis assay was performed on 721.221 HLA-Cw7 target cells expressing mCherry. *P* value was cumulatively calculated for every time point using one-way ANOVA, and Tukeys' post hoc test was used for multiple comparisons, as indicated in the graph (*$P < 0.05$). The analysis was performed for nine individual fields per image (16 images) (with ~$N = 50$ cells in each field). Green line: anergic NS siRNA; red line: anergic Egr2 siRNA; blue line: responsive NS siRNA. Anergic and responsive cells were transfected with EGR2 siRNA or NS siRNA, seeded with 721.221 HLA-Cw7 expressing mCherry cells at an E:T ratio of 10:1, and subjected to Incucyte imaging and analysis; the decrease in fluorescence intensity was measured, and images from the indicated time points are shown. The values were normalized to 721.221 Cherry only—no effector control. Right panel: Images showing cell fluorescence at different time points (scale bar: 200 μm). (**B**) Human CML OTS prepared using K562 CFP cells in Matrigel (1:1–v/v%), were seeded with anergic or responsive pNK cells at an E:T ratio of 5:1. After 6 h, NPs encapsulating Egr2 siRNA or NS siRNA were added to the OTS, and the decrease in fluorescence intensity was monitored and measured; the images along with the fluorescence intensity at the respective time points are shown (numbers on top right of each frame). The decrease in fluorescence over time shows the enhanced cytotoxic activity of anergic cells following Egr2 siRNA treatment (left panel: middle) similar to the responsive cells treated with NS siRNA (left panel: bottom). Whole well images of the OTS at 48 h are presented. Right panel: Images showing cell fluorescence at different time points (Scale bar: 200 μm). (**C**) The tumors were excised on day 27, a single-cell suspension was made, and cells were stained for intracellular Egr2. The pNK cells were distinguished based on FSC vs SSC, and PE content, reflecting nanoparticle incorporation. Representative histogram showing Egr2 expression levels. Blue line shows the EGR2 levels in the group treated with control NP (NS siRNA), and the red line shows the group receiving Egr2 siRNA-NP. (**D**) The dissociated tumors were then subjected to Annexin V staining for apoptotic cells. Representative histogram on the left panel shows the apoptosis of the tumors from each group (one mouse representative of three independent repeats). The right panel shows a graph summarizing the tumors obtained from three different mice. Data are presented as mean ± SEM. *P* values were calculated using one-way ANOVA with Tukeys post hoc test after normalization of the values to the "tumor only" group, which served as the control. Source data are available online for this figure.

