## [Peer Review File · The EMBO Journal]

Dysfunctional natural killer cells can be reprogrammed to regain anti-tumor activity

Batel Sabag, Abhishek Puthenveetil, Moria Levy, Noa Joseph, Tirza Doniger, Orly Yaron, Sarit Karako-Lampert, Itay Lazar, Fatima Awwad, Shahar Ashkenazi, and Mira Barda-Saad

Corresponding author(s): Mira Barda-Saad (Mira.Barda-Saad@biu.ac.il)

Review Timeline:

Submission Date:	1st Oct 23
Editorial Decision:	27th Oct 23
Revision Received:	17th Jan 24
Editorial Decision:	16th Feb 24
Revision Received:	6th Mar 24
Accepted:	8th Mar 24

Editor: Ioannis Papaioannou

Transaction Report:

Dear Prof. Barda-Saad,

Thank you for submitting your manuscript (EMBOJ-2023-115740) for consideration by the EMBO Journal. It has now been seen by three experts in the field, and we have received the full set of their comments, which are included below.

As you will see, the referees mention that the results are novel and significant, they explain that the conclusions of the study are justified, and they commend the solidity of the investigation and the quality of the work. However, ref. #1 also raises the concern of missing controls, and both ref. #1 and #2 provide a number of suggestions for the improvement of the study and the presentation of the results in the manuscript.

Given the referees' positive comments and recommendations, I would like to invite you to submit a revised version of your manuscript, addressing the comments of reviewers #1 and #2. I should add that it is EMBO Journal policy to allow only a single round of major revision, and acceptance of your manuscript will therefore depend on the completeness of your responses in this revised version. If you have any questions or comments, we can also discuss the revisions in a video chat, if you like.

We generally allow three months as standard revision time (26th January 2023). As a matter of policy, competing manuscripts published during this period will not negatively impact our assessment of the conceptual advance presented by your study. However, we request that you contact us as soon as possible upon publication of any related work, to discuss how to proceed. Should you foresee a problem in meeting this three-month deadline, please let us know in advance and we may be able to grant an extension.

Thank you for the opportunity to consider your work for publication in the EMBO Journal. I look forward to your revision.

Yours sincerely,

Instructions for preparing your revised manuscript

1. When you are ready to submit the revision, please upload:

- A Word file of the manuscript text (including legends of main Figures, EV Figures and Tables). Please make sure that changes are highlighted (or "tracked") to be clearly visible.

- Individual production-quality figure files (one file per figure). When assembling your figures, please refer to our figure preparation guidelines in order to ensure proper formatting and readability in print as well as on screen:

If the data shown in a figure are obtained from n {less than or equal to} 2, please use scatter plots showing the individual data points.

- i. the name of the statistical test used to generate error bars and P values
- ii. the number (n) of independent experiments (please specify technical or biological replicates) underlying each data point (discussion of statistical methodology can be reported in the Materials and Methods section, but figure legends should contain a basic description of n , P , and the test applied)
- iii. the nature of the bars and error bars (s.d., s.e.m.).

- A point-by-point response to the referees' comments, with a detailed description of the changes made (as a word file). All referees' concerns must be fully addressed and their suggestions taken on board. When preparing your letter of response to the referees' comments, please bear in mind that this will form part of the Review Process File and will therefore be available online to the community. Please note that you have the possibility to opt out of the transparent process at any stage prior to publication by letting the editorial office know (contact@embojournal.org); if you do opt out, the Review Process File link will point to the following statement: "No Review Process File is available with this article, as the authors have chosen not to make the review process public in this case.". For more details on our Transparent Editorial Process, please visit our website:

<https://www.embopress.org/page/journal/14602075/authorguide#transparentprocess>

- Expanded View (EV) files (replacing Supplementary Information) that are collapsible/expandable online. A maximum of 5 EV Figures can be typeset. EV Figures should be cited as "Figure EV1, Figure EV2" etc. in the text, and their respective legends should be included in the manuscript file after the legends of regular figures. See detailed instructions regarding Expanded View files here:

- For the figures that you do NOT wish to display as Expanded View figures, they should be bundled together with their legends in a single PDF file called "Appendix", which should start with a short Table of Contents (including page numbers). Appendix figures should be referred to in the main text as: "Appendix Figure S1, Appendix Figure S2" etc. Please see detailed instructions here: <https://www.embopress.org/page/journal/14602075/authorguide#expandedview>

- A complete author checklist, which you can download from our author guidelines (<https://www.embopress.org/page/journal/14602075/authorguide>). Please note that the checklist will also be part of the Review Process File.

2. Please note that no statistics should be calculated if $n=2$.

3. Before submitting your revision, primary datasets (and computer code, where appropriate) produced in this study need to be deposited in appropriate public databases (see <https://www.embopress.org/page/journal/14602075/authorguide#dataavailability>). Specifically, we would kindly ask you to provide public access to the following datasets/data:

- RNA sequencing data

The accession numbers and database should be listed in a formal "Data availability" section (placed after Materials and Methods) that follows the model below (see also <https://www.embopress.org/page/journal/14602075/authorguide#dataavailability>):

Data availability

- RNA-seq data: Gene Expression Omnibus GSE46843 (<https://www.ncbi.nlm.nih.gov/geo/query/acc.cgi?acc=GSE46843>)
- [data type]: [name of the resource] [accession number/identifier/doi] ([URL or identifiers.org/DATABASE:ACCESSION])

*** Note: all links should resolve to a page where the data can be accessed. ***

*** Note: the Data Availability Section is restricted to new primary data that are part of this study. ***

4. Please check that the title and the abstract of the manuscript are brief, yet explicit, even to non-specialists. The length of the title should not exceed 100 characters (including spaces), and the abstract should be a single paragraph not exceeding 175 words.

5. Please also note our reference format: <https://www.embopress.org/page/journal/14602075/authorguide#referencesformat>.

7. Please remember: digital image enhancement is acceptable practice, as long as it accurately represents the original data and conforms to community standards. If a figure has been subjected to significant electronic manipulation, this must be noted in the figure legend or in the "Materials and Methods" section. The editors reserve the right to request original versions of figures and the original images that were used to assemble the figure.

8. Our journal encourages inclusion of data citations in the reference list to directly cite datasets that were obtained from public databases. Data citations in the article text are distinct from normal bibliographical citations and should directly link to the database records from which the data can be accessed. In the main text, data citations are formatted as follows: "Data ref: Smith et al, 2001" or "Data ref: NCBI Sequence Read Archive PRJNA342805, 2017". In the Reference list, data citations must be labeled with "[DATASET]". A data reference must provide the database name, accession number/identifiers, and a resolvable link to the landing page from which the data can be accessed at the end of the reference. Further instructions are available at: <https://www.embopress.org/page/journal/14602075/authorguide#referencesformat>.

9. We request authors to consider both actual and perceived competing interests. Please review our policy (<https://www.embopress.org/page/journal/14602075/authorguide#conflictsofinterest>) and update your competing interests

statement if necessary. Please name this section 'Disclosure and competing interests statement' and place it after the Acknowledgements section.

10. Please note that all corresponding authors are required to provide an ORCID ID upon submission of a revised manuscript (<https://orcid.org/>). Please find instructions on how to link your ORCID ID to your account in our manuscript tracking system in our Author guidelines (<https://www.embopress.org/page/journal/14602075/authorguide#authorshipguidelines>).

11. We use CRediT to specify the contributions of each author in the journal submission system. CRediT replaces the author contribution section, which should be removed from the manuscript. Please use the free text box to provide more detailed descriptions. See also guide to authors: <https://www.embopress.org/page/journal/14602075/authorguide#authorshipguidelines>.

13. We would also welcome the submission of cover suggestions or motifs to be used by our Graphics Illustrator in designing a cover.

14. Please use the link below to submit your revision:
<https://emboj.msubmit.net/cgi-bin/main.plex>

Referee #1:

In this manuscript, Sabag et al describe two genes upregulated in anergic/uneducated NK cells, EGR2 and DGKa, and profile their roles in weakening NK cell effector function. Additionally, they demonstrate that these genes are also upregulated in chronically stimulated (exhausted) NK cells in vivo. Using inhibitors and siRNA experiments, they show these proteins inhibit tumor cell lysis. The manuscript puts forward a relatively clear approach and a solid, specific focus on two novel regulators of NK cell function. My suggestions mainly concern how flow data is presented and how certain key in vivo controls are missing for the definiteness of this data.

Major points:

- For much of the flow data throughout the manuscript (i.e. Fig 2), the data are only represented as "relative MFI" comparing expression of markers on anergic and responsive cells. To truly understand the surface expression patterns of these populations, the raw MFI and % of parent should be reported, ideally with an accompanying flow plot (pseudocolor or contour plot). It is very hard to understand the biological relevance of these data without these pieces of information.
- The Fig1 volcano plot does not show EGR2 as indicated in the legend. This is important if it will be a focus of the manuscript as readers need to know fold change and significance.
- In Fig 3 and FigS3, where the tumor-infiltrating NK cells are phenotyped, there is no indication of how many cells are being recovered. I see in Fig S3 that the flow plots begin as pseudocolor dot plots, but then become contour plots as the gated population becomes smaller, which obscures how many individual cells are being analyzed. Without any enumeration of NK cells recovered, it is difficult to interpret the phenotypic data, as the populations being analyzed could be very small. Please include pseudocolor dot plots of your flow data here (e.g. PD-1, EGR2 staining) so we have an idea of cell numbers per sample as well as degree of expression, as the MFI data is relative and not absolute.
- In Fig 3, how are "exhausted" cells being characterized? Are these simply any NK cells recovered from tumor at day 18? Simply because these cells are less functional than naïve pbNK cells does not necessarily mean they are exhausted - these cells are simply "tumor infiltrating NK cells".
- In Fig 5, the Incucyte data need to show more than one single experiment. We need to know if this is reproducible.
- In Fig 5, the tumor study needs to have more control groups - specifically groups of mice that receive tumor + nanoparticle siRNA but no NK cells, to rule out the nanoparticle siRNA influencing the tumor itself. My suggestion would be to use CRISPR to delete these genes from NK cells and then perform the adoptive transfer tumor experiments to show it is having the specific effect on NK cells.
- In Fig 5, blood NK cells and tumor-infiltrating NK cells should be enumerated. Are the differences in tumor growth because there are more NK cells? Or because the NK cells have enhanced function?

Minor points:

- In Fig S1, what is the antigen targeted by the PE-conjugated antibody? It only says "PE" and the legend does not elaborate. Additionally, it would be helpful to see KIR and NKG2A staining separately, ideally in the form of a flow dot plot
- In figure 1D, what does NC mean on the y axis? The legend does not say. Normalized counts?
- In Fig 2, the GSEA data do not have Normalized Enrichment Scores or statistical tests associated with them
- Can the in vivo tumor data be replicated in another tumor model?

Referee #2:

The authors studied an "anergic" subset of NK cells, here defined as lacking KIRs or NKG2A, derived from healthy blood and analyzed via RNA-Seq. The diminished effector functions of these anergic NK cells were compared to responsive NK cells from the same donors. They follow this up with an analysis between their data and the literature comparing anergic with exhausted cells and concluding from this that their studied anergic subset closely resembles the phenotype of viral or tumor exhausted NK cells. Common among this shared phenotype were increased expression of the intrinsic regulators DGKa and Egr2, which the authors independently verified by taking intratumoral "exhausted" NK cells from in vivo mouse models. The effect of these regulators on the anergic/exhausted state were studied via in vitro knockdown assays. This was followed up by studying the negative effects of this pathway on cytotoxicity of the anergic NK cells through in situ and in vivo tumor killing experiments by using a liposomal nanoparticle-based delivery system for Egr2 siRNA. While the authors provide mostly convincing evidence demonstrating the importance of this pathway in NK cell anergy/exhaustion, several issues must be addressed:

1. Why is Figure 2D flipped between responsive and anergy orientations, as compared to Figures 2E and 2F? Also, statistical support for the author's conclusions are required for these GSEA plots, namely normalized enrichment scores (NES) and FDR determinations, which should be < 0.05 . These figures need to be reassessed and better explained, especially with such low enrichment scores.
2. In Figure 2B, the levels (MFI) of DNAM-1, 2B4, and CD160 protein should be compared between anergic and responsive NK cells to be consistent with Figs. 2A and 2C.
3. The authors do not mention the very high expression of KLRC1 (NKG2A) in responsive NK cells, which is marked in Fig 1C. Also, NCAM1 encodes CD56, not CD16 as noted on page 7.

Referee #3:

This manuscript establishes some important regulators of functional and dysfunctional NK cell states which could inform cancer immunotherapy strategies.

The work is of quality and findings well justified

The Mina & Everard Goodman
Faculty of Life Sciences
PROF. MIRA BARDA-SAAD
LABORATORY OF MOLECULAR AND
APPLIED IMMUNOLOGY

January 16th, 2024

Dear Dr. Ioannis Papaioannou,
EMBO Journal

We thank you and the reviewers for your insightful and constructive comments. We have addressed all the points raised by the reviewers, and based on these revisions; we hope that our manuscript will now be found suitable for publication in *The EMBO Journal*.

Reviewer #1:

In this manuscript, Sabag et al describe two genes upregulated in anergic/uneducated NK cells, EGR2 and DGK α , and profile their roles in weakening NK cell effector function. Additionally, they demonstrate that these genes are also upregulated in chronically stimulated (exhausted) NK cells in vivo. Using inhibitors and siRNA experiments, they show these proteins inhibit tumor cell lysis. The manuscript puts forward a relatively clear approach and a solid, specific focus on two novel regulators of NK cell function. My suggestions mainly concern how flow data is presented and how certain key in vivo controls are missing for the definiteness of this data.

Major comments:

1. For much of the flow data throughout the manuscript (i.e. Fig 2), the data are only represented as "relative MFI" comparing expression of markers on anergic and responsive cells. To truly understand the surface expression patterns of these populations, the raw MFI and % of parent should be reported, ideally with an accompanying flow plot (pseudocolor or contour plot). It is very hard to understand the biological relevance of these data without these pieces of information.

A: The anergic sub-population represents approximately 13-20% of the entire NK cell population in the peripheral blood. This limited number of cells available for experimental analysis necessitates the use of bar graphs describing relative MFI values to clearly emphasize the difference between the anergic vs responsive sub-populations. Nevertheless, in response to your concern, we now also provide raw MFI values and the percentage of the parent population into Figure 2, as stipulated. The trends observed within these cohorts exhibit consistency across various analytical representations.

2. The Fig1 volcano plot does not show EGR2 as indicated in the legend. This is important if it will be a focus of the manuscript as readers need to know fold change and significance.

A: We appreciate the comment. The figure legend has been modified accordingly. 3 donors out of 4 exhibited large difference, however, only one donor showed a small difference, in turn influencing the statistical trend. Hence, there were no

significant alterations in EGR2 observed at the RNA level, between the anergic and responsive subsets (Fig. 2G: right panel). However, the protein level exhibited a noteworthy elevation in Egr2 of approximately 3.5 fold in anergic NK cells, as shown consistently from 10 donors (Fig. 2H).

3. In Fig3 and FigS3, where the tumor-infiltrating NK cells are phenotyped, there is no indication of how many cells are being recovered. I see in Fig S3 that the flow plots begin as pseudocolor dot plots, but then become contour plots as the gated population becomes smaller, which obscures how many individual cells are being analyzed. Without any enumeration of NK cells recovered, it is difficult to interpret the phenotypic data, as the populations being analyzed could be very small.

A: As previously demonstrated (PMID: 33299656), natural killer (NK) cells are predominantly absent from tumor tissues, constituting only 5-12% of the total cell population. However, a limited subset of NK cells is present within the tumor site. Despite their relatively low abundance, NK cells have been implicated in playing a potential role in disease control (PMIDs 30911118 and 3615679). We agree with your comment and have therefore included enumeration expressed as percentages for each re-gating – 10.8% of NK cells, 97.6% singlets, and the CD45+ population (95.9%) recovered for further downstream analysis and now represented in pseudocolor dot plots.

Please include pseudocolor dot plots of your flow data here (e.g. PD-1, EGR2 staining) so we have an idea of cell numbers per sample as well as degree of expression, as the MFI data is relative and not absolute.

A: The contour plots enumerating the percentage of primary human NK cells expressing CD107a, PD-1, DKG α and EGR2 are now included in the Expanded View (EV) Fig. 3, next to the respective histograms.

4. In Fig 3, how are "exhausted" cells being characterized? Are these simply any NK cells recovered from tumor at day 18? Simply because these cells are less functional than naïve pbNK cells does not necessarily mean they are exhausted - these cells are simply "tumor infiltrating NK cells".

A: The text has been modified and the NK cells from the tumor are now referred to as tumor infiltrating NK cells (TINK), representing a dysfunctional – 'exhausted' phenotype characterized by reduced degranulation (CD107a expression) and upregulated PD-1 expression. Additionally, the NK cells from the tumors were excised on day 18, a time point that was shown in Fig. 3B, to represent a plateau of tumor growth, indicating the inability of NK cells to restrain tumor progression.

5. In Fig 5, the Incucyte data need to show more than one single experiment. We need to know if this is reproducible.

A: Based on this suggestion, we now show all three repeats overlaid onto a single graph, Fig. 5C, and the previous figure has been replaced with an updated version.

Rebuttal Figure 1: (A) Nanoparticle specificity. Gating strategy and the efficiency of EGR2 gene silencing. **(B) Human Protein Atlas Database** NKp46 and EGR2 expression in PANC-1 cell lines **(C)** Internal 'in house' control (green line) for in vivo experiments – Fig 5.

6. In Fig 5, the tumor study needs to have more control groups - specifically groups

of mice that receive tumor + nanoparticle siRNA but no NK cells, to rule out the nanoparticle siRNA influencing the tumor itself.

A: Based on this valid concern, the effect of the Egr2 siRNA on the tumor cells was determined in the same experiment (Rebuttal Fig. 1C, Fig 5D-E). Following tumor extraction and subsequent single-cell dissociation, we conducted *ex vivo* FACS analysis. Notably, the NPs are labeled with Rhodamine, rendering them detectable by FACS and enabling the measurement of nanoparticle uptake by tumor cells. 14% of tumor cells exhibited rhodamine staining reflecting non-specific NP uptake (Rebuttal Figure 1A). Although PANC-1 cell lines express EGR2 they lack NKp46 expression for efficient NP uptake (The Human Protein Atlas (Rebuttal Figure 1B)). Despite the 14% non-specific uptake of the nanoparticles in PANC-1 tumor cells, the encapsulated siRNA did not affect Egr2 expression. This suggests that Panc-1 cells lacking NKp46 expression are unable to internalize encapsulated siRNA. In contrast, NK cells, facilitated by nanoparticles tagged with anti-NKp46, recognize NKp46 on intra-tumor pNK, leading to the cytoplasmic release of siRNA via receptor-mediated endocytosis. Intriguingly, intra-tumor NK cells exhibited a noteworthy 97.9% nanoparticle uptake and a significant reduction in Egr2 expression (Rebuttal Fig. 1A). Additionally, there was a striking increase in side scatter (SSC-A), suggesting enhanced granularity following Egr2 silencing, as observed in our in-vivo experiments. We include the corresponding figures here, showing the outcomes with both untreated tumor and tumors treated with nanoparticles. Additionally, we provide here a comprehensive figure that includes the results of an internal control group consisting of four mice (Rebuttal Fig. 1C - green lines). The data was not shown since it was an internal control and not the focus of the manuscript. This group received NP-encapsulated NS siRNA; notably, this treatment did not significantly impact tumor growth when compared to the tumor-only control group. These results support that tumor growth rate is affected by the primary NK cells taking the nanoparticles encapsulating EGR2 siRNA and not the specific effects of the nanoparticles on the tumor itself. Thus, this did not influence the conclusions of our experiment.

My suggestion would be to use CRISPR to delete these genes from NK cells and then perform the adoptive transfer tumor experiments to show it is having the specific effect on NK cells.

A: Egr2, a protein with pivotal significance in immune cells, plays a critical role, and the knockout of its gene has been shown to adversely impact cellular activity (PMIDs 23021953 and 28487311). Consequently, our objective was to downregulate the expression of Egr2, specifically in phenotypically exhausted tumor-infiltrating NK cells. Additionally, the process of adoptive transfer of NK cells often involves ex-vivo expansion, which can lead to the over-activation of these cells. It is crucial to emphasize that our primary objective was to employ the nanoparticle-based drug delivery platform as a means to circumvent the need for adoptive transfer and to directly modulate NK cell functions in their native in vivo environment. This strategy not only avoids potential harm to primary cells, but also enables more precise and physiologically relevant modulation of NK cell behavior.

7. In Fig 5, blood NK cells and tumor-infiltrating NK cells should be enumerated. Are the differences in tumor growth because there are more NK cells? Or because the NK cells have enhanced function?

A: Thank you for this excellent suggestion. It is evident that the cells in the group receiving EGR2 siRNA exhibit significantly higher functionality, as indicated by elevated degranulation levels. We conducted a quantification of tumor-infiltrating NK cells in both the NS siRNA and EGR2 siRNA groups. Importantly, we observed no statistically significant difference between the two groups, suggesting that the primary factor contributing to enhanced functionality is not the quantity of NK cells but rather an intrinsic improvement in their functional attributes. The graph "Rebuttal Figure 2" represents the percentage of NK cells extracted from the tumors, along with the accompanying statistical analysis, which is provided here.

Rebuttal Figure 2: Percentage of NK cells obtained from the tumors on day 27. n=4, P=0.1075.

Minor points:

1. In Fig S1, what is the antigen targeted by the PE-conjugated antibody? It only says "PE" and the legend do not elaborate. Additionally, it would be helpful to see KIR and NKG2A staining separately, ideally in the form of a flow dot plot.

A: The legend now provides more detail describing the PE-conjugated antibodies. The sorting process utilizes a panel of PE-conjugated antibodies targeting both pan-KIR (KIR2D and KIR3DL1) and NKG2A. We opted for a protocol that could distinguish between cells that express none of the classical inhibitory receptors versus the group that expresses these receptors, even at very low levels; choosing a single fluorophore for all the receptors allowed us to enhance the signal for each cell. Furthermore, the decision to limit the sorting to a single laser channel was deliberate, and aimed at enhancing both the efficiency and purity of the sorting process. Moreover, in consideration of downstream experiments that involve FACS analysis, this approach allows for flexibility in the selection of additional fluorophores within the experimental panel.

2. In figure 1D, what does NC mean on the y axis? The legend does not say. Normalized counts?

A: "NC" is now defined in the legend as normalized counts.

3. In Fig 2, the GSEA data do not have Normalized Enrichment Scores or statistical tests associated with them

A: Scores and statistics are now included in the legend.

4. Can the in vivo tumor data be replicated in another tumor model?

A: We expect that this approach would be applicable across a range of tumor models. There is a consistent pattern of dysfunction in different solid tumor types.

Reviewer #2:

The authors studied an "anergic" subset of NK cells, here defined as lacking KIRs or NKG2A, derived from healthy blood and analyzed via RNA-Seq. The diminished effector functions of these anergic NK cells were compared to responsive NK cells from the same donors. They follow this up with an analysis between their data and the literature comparing anergic with exhausted cells and concluding from this that their studied anergic subset closely resembles the phenotype of viral or tumor exhausted NK cells. Common among this shared phenotype were increased expression of the intrinsic regulators DGK α and Egr2, which the authors independently verified by taking intratumoral "exhausted" NK cells from in vivo mouse models. The effect of these regulators on the anergic/exhausted state was studied via in vitro knockdown assays. This was followed up by studying the negative effects of this pathway on cytotoxicity of the anergic NK cells through in situ and in vivo tumor killing experiments by using a liposomal nanoparticle-based delivery system for Egr2 siRNA. While the authors provide mostly convincing evidence demonstrating the importance of this pathway in NK cell anergy/exhaustion, several issues must be addressed:

Major comments:

1. *Why is Figure 2D flipped between responsive and anergy orientations, as compared to Figures 2E and 2F? Also, statistical support for the author's conclusions are required for these GSEA plots, namely normalized enrichment scores (NES) and FDR determinations, which should be < 0.05. These figures need to be reassessed and better explained, especially with such low enrichment scores.*

A: Thank you for your valuable comment. The statistical values, NES and FDR q-values are now included in the figure and the associated figure legend. Further, the text in the results section has been modified accordingly. Fig. 2D is now flipped between the anergic and responsive orientation.

2. *In Figure 2B, the levels (MFI) of DNAM-1, 2B4, and CD160 protein should be compared between anergic and responsive NK cells to be consistent with Figs. 2A and 2C.*

A: To ensure consistency across the figures, we have now included the protein level data of DNAM-1, 2B4 and CD160. The updated figure (2B) is now included in the revised manuscript.

3. *The authors do not mention the very high expression of KLRC1 (NKG2A) in responsive NK cells, which is marked in Fig 1C.*

A: The high expression of NKG2A in responsive NK cells reflects the efficiency of our sorting of the anergic versus the responsive NK cell subsets. NKG2A is essential for acquiring functional responses and NK cell education, although it is an inhibitory receptor (PMID: 33101277, PMID: 20818413, PMID: 18974374, PMID: 31676749). These considerations are now explained in the manuscript.

Also, NCAM1 encodes CD56, not CD16 as noted on page 7.

A: Thank you for this correction. This statement has now been corrected.

Reviewer #3:

This manuscript establishes some important regulators of functional and dysfunctional NK cell states which could inform cancer immunotherapy strategies.

The work is of quality and findings well justified.

A: Thank you.

Dear Prof. Barda-Saad,

Thank you for the submission of your revised manuscript to The EMBO Journal. We have now received the comments of the two referees that were asked to re-assess your study (included below). Both referees are satisfied with the revision, but there is a remaining concern of referee #2 that we need you to address in a revised version of your manuscript before we can accept it for publication. The referee points out that the data presented in Figure panels 2D, E, and F are difficult to understand and should either be improved and explained better, or removed from the manuscript if they are not conclusive enough. Please see the detailed comment of the referee below and revise your manuscript accordingly.

From the editorial side, there are also a few minor changes or corrections that we need from you before we can proceed with acceptance of the manuscript:

- You have included two different lists of keywords in your manuscript. Please merge these two lists into one list of up to 5 short keywords, after the Abstract.
- Please make sure that the deposited data will be publicly available at the time of publication. The reviewer access code can now be removed from the Data availability section.
- Please change the heading of your conflict-of-interest statement to "Disclosure and competing interests statement".
- The funding information should be included in the Acknowledgements section.
- The author contributions statement should be removed from the manuscript file. Instead, we now use CRediT to specify the contributions of each author in the journal submission system. Please use the free text box to provide more detailed descriptions during submission. See also our guide to authors for more information:
<https://www.embopress.org/page/journal/14602075/authorguide#authorshipguidelines>.
- We noticed that there is a callout for Fig. 1F, but no such panel exists in Fig. 1. Please check and correct, or remove.
- The Appendix Table S1 is not called out in the manuscript; please make sure that all Appendix items are called out in the main manuscript file.
- The manuscript ID number is missing in the general info table of your Author Checklist (at the top of the sheet). Please include it in a revised checklist.
- Please include in the Materials and Methods the reference number of approval of your experiments involving human participants by the Bar-Ilan University Ethics committee (in the paragraph "PBMC isolation").
- The table in the Materials and Methods should either be called "Table 1", moved to the bottom of the manuscript file (or uploaded separately), and called out as appropriate in the text, or the information included in it should be incorporated into the respective paragraph of the Materials and Methods. Please see our guide to authors for more information:
<https://www.embopress.org/page/journal/14602075/authorguide#tablesformat>.
- Please note that a number of changes are necessary in your Appendix:
 1. Please include a brief Table of Contents with page numbers on its first page.
 2. The nomenclature of the items included in the Appendix should be corrected to "Appendix Figure S1" and "Appendix Table S1" (please update their callouts accordingly throughout the Appendix and the manuscript file).
 3. Their legends/captions should appear directly above or below them in the Appendix.
 4. The references should be moved to the main References list of your manuscript.
 5. Please remove the Appendix description from the bottom of your manuscript file.
- Please also note that the Table included in your Appendix could alternatively be formatted as a "Reagent and Tools Table" and included in the main manuscript file if you prefer to use the Structured Methods format for your article. Please see our guide for more information, examples/templates and instructions:
<https://www.embopress.org/page/journal/14602075/authorguide#researcharticleguide>.
- The Expanded View Source Data file needs to be zipped.
- Please note that EMBO press papers are accompanied online by:
 - A) a short (2 sentences) summary of the findings and their significance,
 - B) 2-4 short bullet points highlighting the key results, and
 - C) a synopsis image in .jpg or .png format that is exactly 550 pixels wide and 300-600 pixels high (the height is variable). You can either show a model or key data in the synopsis image. Please note that the text needs to be legible at the final size.

Please upload this information along with your revised manuscript (the text for A and B should be provided in a separate Word file).

- Please note that a separate "Data Information" section is required in the legends of Figures 2a-c, h; 3b-f. See our guide for more information and an example: <https://www.embopress.org/page/journal/14602075/authorguide#figureformat>.
- The legends of Expanded View Figures 3-5 are labelled as supplementary data 3-5 in the manuscript. This needs to be rectified.
- Please define the annotated p values ***/**/* in the legends of Figures 3b; 5c; EV 5a; as appropriate.
- Please indicate the statistical test used for data analysis in the legends of Figures 1c; 5a-b, f; EV 1d-g.
- Please note that information related to "n" is missing in the legends of Figures 1c; 3b; 5f, h-i.
- Although "n" is provided, please describe the nature of entity for "n" in the legends of Figures 2c, g; 3e, g-i; EV 1b-c; EV 2e-f; EV 4a, c-e.
- Please note that the error bars are not defined in the legends of Figures 5f-g; EV 4a.
- Please note that the scale bar needs to be defined in the legend of Figure EV 5b.

As soon as these issues are resolved, I will contact you again to discuss with you a few suggestions for minor textual improvements in the title and abstract.

Please also note that as part of the EMBO publications' Transparent Editorial Process, The EMBO Journal publishes online a Peer Review File along with each accepted manuscript. This File will be published in conjunction with your paper and will include the referee reports, your point-by-point response and all pertinent correspondence relating to the manuscript. You can opt out of this by letting the editorial office know (contact@embojournal.org). If you do opt out, the Peer Review File link will point to the following statement: "No Peer Review File is available with this article, as the authors have chosen not to make the review process public in this case."

We look forward to seeing a final version of your manuscript as soon as possible. Please use this link to submit your revision: <https://emboj.msubmit.net/cgi-bin/main.plex>

Yours sincerely,

Referee #1:

The manuscript is improved and my queries have been addressed to improve the quality of this manuscript.

Referee #2:

I am not an expert in GSEA plots, but I am still having a hard time wrapping my head around the data presented in Figures 2D,E,F and how these comparisons were performed and match the authors' conclusions. These either need to be better explained or some of the plots should be removed from the manuscript. First, I do not understand why expression of genes in the hepatitis B patients are coordinately increased in the authors' anergic cell dataset in Figure 2D, whereas those increased in their anergic cell dataset appear to be increased in the healthy donor dataset in Figures 2E and 2F (not with the tumor infiltrating lymphocyte or viral infection related genes, as would be expected). Second, a FDR value of 0.79 for the data in Figure 2E is far from statistically significant and this plot should be removed. The GSEA plots in Figure EV2A,B have the best FDR values and seem to better support the conclusions. Perhaps these would be better to include in the main figure. The authors should consult

with a bioinformatician with experience in GSEA analyses and only include data that clearly support their conclusions. All other aspects of the manuscript are of high quality, but these panels need to be either removed, performed in a different manner, or better explained to justify their interpretation.

The Mina & Everard Goodman
Faculty of Life Sciences
PROF. MIRA BARDASAAD
LABORATORY OF MOLECULAR AND
APPLIED IMMUNOLOGY

March 6th, 2024
Dr. Ioannis Papaioannou,
The EMBO Journal

Dear Dr. Ioannis Papaioannou,
We thank you and the reviewers for your insightful and constructive comments. We have addressed all the points raised by the reviewers and based on these revisions; we hope that our manuscript will now be found suitable for publication in *The EMBO Journal*.

Referee #1:

The manuscript has been improved, and my queries have been addressed to improve the quality of this manuscript.

We thank the reviewer for critically reviewing our manuscript.

Referee #2:

I am not an expert in GSEA plots, but I am still having a hard time wrapping my head around the data presented in Figures 2D, E, F and how these comparisons were performed and match the authors' conclusions. These either need to be better explained or some of the plots should be removed from the manuscript. First, I do not understand why expression of genes in the hepatitis B patients are coordinately increased in the authors' anergic cell dataset in Figure 2D, whereas those increased in their anergic cell dataset appear to be increased in the healthy donor dataset in Figures 2E and 2F (not with the tumor infiltrating lymphocyte or viral infection related genes, as would be expected). Second, a FDR value of 0.79 for the data in Figure 2E is far from statistically significant and this plot should be removed. The GSEA plots in Figure EV2A, B have the best FDR values and seem to better support the conclusions. Perhaps these would be better to include in the main figure. The authors should consult with a bioinformatician with experience in GSEA analyses and only include data that clearly support their conclusions. All other aspects of the manuscript are of high quality, but these panels need to be either removed, performed in a different manner, or better explained to justify their interpretation.

We appreciate the reviewer's comment. We would like to emphasize that the analysis of the RNA seq and specifically GSEA data was performed by the bioinformatic unit of Bar-Ilan University. The GSEA plot (Fig. 2E) has been removed and Fig. EV2A and B have been moved to the main manuscript file and are now Fig. 2F and G. A negative enrichment score indicates that the genes within the gene set are more enriched in the lower-ranked portion of the list (Enrichment column in the source data file indicates the enriched genes in the anergic subset).

Dear Prof. Barda-Saad,

I am very pleased to inform you that your manuscript has been accepted for publication in The EMBO Journal.

Yours sincerely,
